# Indiscriminate Data Poisoning Attacks on Neural Networks*

**Yiwei Lu**                                                    *yiwei.lu@uwaterloo.ca*
*University of Waterloo*

**Gautam Kamath**†                                              *g@csail.mit.edu*
*University of Waterloo*

**Yaoliang Yu**‡                                                *yaoliang.yu@uwaterloo.ca*
*University of Waterloo*

**Reviewed on OpenReview:** *https://openreview.net/forum?id=x4hmIsWu7e*

## Abstract

Data poisoning attacks, in which a malicious adversary aims to influence a model by injecting "poisoned" data into the training process, have attracted significant recent attention. In this work, we take a closer look at existing poisoning attacks and connect them with old and new algorithms for solving sequential Stackelberg games. By choosing an appropriate loss function for the attacker and optimizing with algorithms that exploit second-order information, we design poisoning attacks that are effective on neural networks. We present efficient implementations by parameterizing the attacker and allowing simultaneous and coordinated generation of tens of thousands of poisoned points, in contrast to most existing methods that generate poisoned points one by one. We further perform extensive experiments that empirically explore the effect of data poisoning attacks on deep neural networks. Our paper sets a new benchmark on the possibility of performing indiscriminate data poisoning attacks on modern neural networks.

## 1 Introduction

Adversarial attacks have repeatedly exposed critical vulnerabilities in modern machine learning (ML) models (Nelson et al., 2008; Szegedy et al., 2013; Kumar et al., 2020). As ML systems are deployed in increasingly important settings, significant effort has been levied in understanding attacks and defenses towards *robust* machine learning.

In this paper, we focus on *data poisoning attacks*. ML models require a large amount of data to achieve good performance, and thus practitioners frequently gather data by scraping content from the web (Gao et al., 2020; Wakefield, 2016). This gives rise to an attack vector, in which an adversary may manipulate part of the training data by injecting poisoned samples. For example, an attacker can *actively* manipulate datasets by sending corrupted samples directly to a dataset aggregator such as a chatbot, a spam filter, or user profile databases; the attacker can also *passively* manipulate datasets by placing poisoned data on the web and waiting for collection. Moreover, in *federated learning*, adversaries can also inject malicious data into a diffuse network (Shejwalkar et al., 2021; Lyu et al., 2020).

A spectrum of such data poisoning attacks exists in the literature, including *targeted*, *indiscriminate* and *backdoor* attacks. We focus on indiscriminate attacks for image classification, where the attacker aims at decreasing the overall test accuracy of a model by adding a small portion of poisoned points. Current indiscriminate attacks are most effective against convex models (Biggio et al., 2011; Koh & Liang, 2017;

---

*GK and YY are listed in alphabetical order.

†Supported by an NSERC Discovery Grant, an unrestricted gift from Google, and a University of Waterloo startup grant.

‡Supported by an NSERC Discovery Grant, Canada CIFAR AI chair program and WHJIL.

Koh et al., 2018; Shumailov et al., 2021), and several defenses have also been proposed (Steinhardt et al., 2017; Diakonikolas et al., 2019). However, existing poisoning attacks are less adequate against more complex non-convex models, especially deep neural networks, either due to their formulation being inherently tied to convexity or computational limitations. For example, many prior attacks generate poisoned points sequentially. Thus, when applied to deep models or large datasets, these attacks quickly become computationally infeasible. To our knowledge, a systematic analysis of indiscriminate data poisoning attacks on deep neural works is still largely missing—a gap we aim to fill in this work.

To address this difficult problem, we design more versatile data poisoning attacks by formulating the problem as a non-zero-sum Stackelberg game, in which the attacker crafts some poisoned points with the aim of decreasing the test accuracy, while the defender optimizes its model on the poisoned training set. We exploit second-order information and apply the Total Gradient Descent Ascent (TGDA) algorithm to address the attacker's objective, even on non-convex models.

We also examine the effectiveness of alternative formulations, including the simpler zero-sum setting as well as when the defender leads the optimization. Moreover, we address computational challenges by proposing an efficient architecture for poisoning attacks, where we parameterize the attacker as a separate network rather than optimizing the poisoned points directly. By applying TGDA to update the attacker model directly, we are able to generate tens of thousands of poisoned points simultaneously in one pass, potentially even in a coordinated way.

In this work, we make the following contributions:

- We construct a new data poisoning attack based on TGDA that incorporates second-order optimization. In comparison to prior data poisoning attacks, ours is significantly more effective and runs at least an order of magnitude faster.
- We summarize and classify existing data poisoning attacks (specifically, indiscriminate attacks) in both theoretical formulations and experimental settings.
- We propose an efficient attack architecture, which enables a more efficient clean-label attack.
- We conduct experiments to demonstrate the effectiveness of our attack on neural networks and its advantages over previous methods.

**Notation.** Throughout this paper, we denote training data as $\mathcal{D}_{tr}$, validation data as $\mathcal{D}_v$, test data as $\mathcal{D}_{test}$, and poisoned data as $\mathcal{D}_p$. We use $\mathsf{L}$ to denote the leader in a Stackelberg game, $\ell$ for its loss function, $\mathbf{x}$ for its action, and $\theta$ for its model parameters (if they exist). Similarly, we use $\mathsf{F}$ to denote the follower, $f$ for its loss function, and $\mathbf{w}$ for its model parameters. Finally, we use $\varepsilon$ as the poison budget, namely that $|\mathcal{D}_p| = \varepsilon |\mathcal{D}_{tr}|$.

## 2 Background

In this section, we categorize existing data poisoning attacks according to the attacker's *power* and *objectives*, and specify the type of attack we study in this paper.

### 2.1 Power of an attacker

**Injecting poisoned samples.** Normally, without breaching the defender's database (i.e., changing the existing training data $\mathcal{D}_{tr}$), an attacker can only *inject* poisoned data, actively or passively to the defender's database, such that its objective can be achieved when the model is retrained after collection. Such a situation may be realistic when the defender gathers data from several sources, some of which may be untrusted (e.g., when scraping data from the public Internet). The goal of the attacker can be presented as:

$$\mathbf{w}_* = \mathbf{w}_*(\mathcal{D}_p) \in \arg\min_{\mathbf{w}} \ \mathcal{L}(\mathcal{D}_{tr} \cup \mathcal{D}_p, \mathbf{w}), \tag{1}$$

where $\mathbf{w}_*$ is the attacker's desired model parameter, which realizes the attacker's objectives, $\mathcal{L}(\cdot)$ is the loss function. We focus on such attacks and further categorize them in the next subsection.

**Perturbing training data.** Some work makes the assumption that the attacker can directly change the training data $\mathcal{D}_{tr}$. This is perhaps less realistic, as it assumes the attacker has compromised the defender's database. We note that this threat model may be more applicable in an alternate setting, where the defender wishes to prevent the data from being used downstream to train a machine learning model. This research direction focuses on so called *unlearnable examples* (Huang et al., 2021; Yu et al., 2021; Fowl et al., 2021b;a), and has faced some criticism that it provides "a false sense of security" (Radiya-Dixit et al., 2022). We provide more details of this line of research in Appendix B.

In this paper, we focus on injecting poisoned samples as it is a more realistic attack.

## 2.2 Objective of an attacker

Data poisoning attacks can be further classified into three categories according to the adversary's objective (Cinà et al., 2022; Goldblum et al., 2022).

**Targeted attack.** The attacker adds poisoned data $\mathcal{D}_p$ resulting in a $\mathbf{w}^*$ such that a particular target example from the test set is misclassified as the *base* class (Shafahi et al., 2018; Aghakhani et al., 2020; Guo & Liu, 2020; Zhu et al., 2019). This topic is well studied in the literature, and we refer the reader to Schwarzschild et al. (2021) for an excellent summary of existing methods.

**Backdoor attack.** This attack aims at misclassifying any test input with a particular trigger pattern (Gu et al., 2017; Tran et al., 2018; Chen et al., 2018; Saha et al., 2020). Note that backdoor attacks require access to both the training data as well as the input at inference time to plant the trigger.

**Indiscriminate attack.** This attack aims to induce a parameter vector $\mathbf{w}^*$ that broadly decreases the model utility. We consider image classification tasks where the attacker aims to reduce the overall classification accuracy. Existing methods make different assumptions on the attacker's knowledge:

- Perfect knowledge attack: the attacker has access to both training and test data ($\mathcal{D}_{tr}$ and $\mathcal{D}_{test}$), the target model, and the training procedure (e.g., the min-max attack of Koh et al. 2018).
- Training-only attack: the attacker has access to training data $\mathcal{D}_{tr}$, the target model, and the training procedure (e.g., Muñoz-González et al. 2017; Biggio et al. 2012).
- Training-data-only attack: the attacker only has access to the training data $\mathcal{D}_{tr}$ (e.g., the label flip attack of Biggio et al. 2011).

In Appendix A we give a more detailed summary of the existing indiscriminate data poisoning attacks.

In this work, we focus on training-only attacks because perfect knowledge attacks are not always feasible due to the proprietary nature of the test data, while existing training-data-only attacks are weak and often fail for deep neural networks, as we show in Section 5.

## 3 Total Gradient Descent Ascent Attack

In this section, we formulate the indiscriminate attack and introduce our attack algorithm. We first briefly introduce the Stackelberg game and then link it to data poisoning.

### 3.1 Preliminaries on Stackelberg Game

The Stackelberg competition is a strategic game in Economics in which two parties move sequentially (von Stackelberg, 1934). Specifically, we consider two players, a leader $\mathsf{L}$ and a follower $\mathsf{F}$ in a Stackelberg game, where the follower $\mathsf{F}$ chooses $\mathbf{w}$ to best respond to the action $\mathbf{x}$ of the leader $\mathsf{L}$, through minimizing its loss function $f$:

$$\forall \mathbf{x} \in \mathbb{X} \subseteq \mathbb{R}^d, \quad \mathbf{w}_*(\mathbf{x}) \in \operatorname*{arg\,min}_{\mathbf{w} \in \mathbb{W}} f(\mathbf{x}, \mathbf{w}), \tag{2}$$

and the leader $\mathsf{L}$ chooses $\mathbf{x}$ to maximize its loss function $\ell$:

$$\mathbf{x}_* \in \operatorname*{arg\,max}_{\mathbf{x} \in \mathbb{X}} \ell(\mathbf{x}, \mathbf{w}_*(\mathbf{x})), \tag{3}$$

where $(\mathbf{x}_*, \mathbf{w}_*(\mathbf{x}_*))$ is known as a Stackelberg equilibrium. We note that an early work of Liu & Chawla (2009) already applied the Stackelberg game formulation to learning a linear discriminant function, where an adversary perturbs the whole training set first. In contrast, we consider the poisoning problem where the adversary can only add a small amount of poisoned data to the *unperturbed* training set. Moreover, instead of the genetic algorithm in Liu & Chawla (2009), we solve the resulting Stackelberg game using gradient algorithms that are more appropriate for neural network models. The follow-up work of Liu & Chawla (2010) further considered a constant-sum simplification, effectively crafting unlearnable examples (see more discussion in Appendix B) for support vector machines and logistic regression. Finally, we mention the early work of Dalvi et al. (2004), who essentially considered a game-theoretic formulation of adversarial training. However, the formulation of Dalvi et al. (2004) relied on the notion of Nash equilibrium where both players move simultaneously while in their implementation the attacker perturbs the whole training set w.r.t a fixed surrogate model (naive Bayes).

When $f = \ell$ we recover the zero-sum setting where the problem can be written compactly as:

$$\max_{\mathbf{x} \in \mathbb{X}} \min_{\mathbf{w} \in \mathbb{W}} \ell(\mathbf{x}, \mathbf{w}), \tag{4}$$

see, e.g., Zhang et al. (2021) and the references therein.

For simplicity, we assume $\mathbb{W} = \mathbb{R}^p$ and the functions $f$ and $\ell$ are smooth, hence the follower problem is an instance of unconstrained smooth minimization.

## 3.2 On Data Poisoning Attacks

There are two possible ways to formulate data poisoning as a Stackelberg game, according to the acting order. Here we assume the attacker is the leader and acts first, and the defender is the follower. This assumption can be easily reversed such that the defender acts first. Both of these settings are realistic depending on the defender's awareness of data poisoning attacks. We will show in Section 5 that the ordering of the two parties affects the results significantly.

**Non-zero-sum formulation.** In this section, we only consider the attacker as the leader as the other case is analogous. Here recall that the follower $\mathsf{F}$ (i.e., the defender) aims at minimizing its loss function $f = \mathcal{L}(\mathcal{D}_{tr} \cup \mathcal{D}_p, \mathbf{w})$ under data poisoning:

$$\mathbf{w}_* = \mathbf{w}_*(\mathcal{D}_p) \in \arg \min_{\mathbf{w}} \ \mathcal{L}(\mathcal{D}_{tr} \cup \mathcal{D}_p, \mathbf{w}), \tag{5}$$

while the leader $\mathsf{L}$ (i.e., the attacker) aims at maximizing a different loss function $\ell = \mathcal{L}(\mathcal{D}_v, \mathbf{w}_*)$ on the validation set $\mathcal{D}_v$:

$$\mathcal{D}_{p_*} \in \arg \max_{\mathcal{D}_p} \mathcal{L}(\mathcal{D}_v, \mathbf{w}_*), \tag{6}$$

where the loss function $\mathcal{L}(\cdot)$ can be any task-dependent target criterion, e.g., the cross-entropy loss. Thus we have arrived at the following non-zero-sum Stackelberg formulation of data poisoning attacks (a.k.a., a bilevel optimization problem, see e.g. Muñoz-González et al. 2017; Huang et al. 2020; Koh et al. 2018):

$$\max_{\mathcal{D}_p} \mathcal{L}(\mathcal{D}_v, \mathbf{w}_*), \ \text{s.t.} \ \mathbf{w}_* \in \arg \min_{\mathbf{w}} \mathcal{L}(\mathcal{D}_{tr} \cup \mathcal{D}_p, \mathbf{w}). \tag{7}$$

Note that we assume that the attacker can inject $\varepsilon N$ poisoned points, where $N = |\mathcal{D}_{tr}|$ and $\varepsilon$ is the power of the attacker, measured as a fraction of the training set size.

We identify that Equation (7) is closely related to the formulation of unlearnable examples (Liu & Chawla, 2010; Huang et al., 2021; Yu et al., 2021; Fowl et al., 2021b;a; Sandoval-Segura et al., 2022; Fu et al., 2021):

$$\max_{\mathcal{D}_p} \mathcal{L}(\mathcal{D}_v, \mathbf{w}_*), \ \text{s.t.} \ \mathbf{w}_* \in \arg \min_{\mathbf{w}} \mathcal{L}(\mathcal{D}_p, \mathbf{w}). \tag{8}$$

where $\mathcal{D}_p = \{(x_i + \sigma_i, y_i)\}_{i=1}^N$, $\sigma_i$ is the bounded sample-wise or class-wise perturbation ($\|\sigma_i\|_p \le \varepsilon_\sigma$). The main differences lie in the direct modification of the training set $\mathcal{D}_{tr}$ (often all of it). In comparison, adding poisoned points to a clean training set would never results in 100 % modification in the augmented training set. This seemingly minor difference can cause a significant difference in algorithm design and performance. We direct interested readers to Appendix B for details.

**Previous approaches.** Next, we mention three previous approaches for solving Equation (7).

(1) A direct approach: While the inner minimization can be solved via gradient descent, the outer maximization problem is non-trivial as the dependence of $\mathcal{L}(\mathcal{D}_v, \mathbf{w}_*)$ on $\mathcal{D}_p$ is *indirectly* through the parameter $\mathbf{w}$ of the poisoned model. Thus, *applying simple algorithms (e.g., Gradient Descent Ascent) directly would result in zero gradients in practice.* Nevertheless, we can rewrite the desired derivative using the chain rule:

$$\frac{\partial \mathcal{L}(\mathcal{D}_v, \mathbf{w}_*)}{\partial \mathcal{D}_p} = \frac{\partial \mathcal{L}(\mathcal{D}_v, \mathbf{w}_*)}{\partial \mathbf{w}_*} \frac{\partial \mathbf{w}_*}{\partial \mathcal{D}_p}. \tag{9}$$

The difficulty lies in computing $\frac{\partial \mathbf{w}_*}{\partial \mathcal{D}_p}$, i.e., measuring how much the model parameter $\mathbf{w}$ changes with respect to the poisoned points $\mathcal{D}_p$. Biggio et al. (2011) and Koh & Liang (2017) compute $\frac{\partial \mathbf{w}_*}{\partial \mathcal{D}_p}$ exactly via KKT conditions while Muñoz-González et al. (2017) approximate it using gradient ascent. We characterize that the Back-gradient attack in Muñoz-González et al. (2017) can be understood as the k-step unrolled gradient descent ascent (UGDA) algorithm in Metz et al. (2017):

$$\mathbf{x}_{t+1} = \mathbf{x}_t + \eta_t \nabla_{\mathbf{x}} \ell^{[k]}(\mathbf{x}_t, \tilde{\mathbf{w}}_t), \tag{10}$$
$$\mathbf{w}_{t+1} = \mathbf{w}_t - \eta_t \nabla_{\mathbf{w}} f(\mathbf{x}_t, \mathbf{w}_t) \tag{11}$$

where $\ell^{[k]}(\mathbf{x}, \mathbf{w})$ is the $k$-time composition, i.e., we perform $k$ steps of gradient descent for the leader. Furthermore, Huang et al. (2020) propose to use a meta-learning algorithm for solving a similar bilevel optimization problem in targeted attack, and can be understood as running UGDA for $M$ models and taking the average.

(2) Zero-sum reduction: Koh et al. (2018) also proposed a reduced problem of Equation (7), where the leader and follower share the same loss function (i.e. $f = \ell$):

$$\max_{\mathcal{D}_p} \min_{\mathbf{w}} \mathcal{L}(\mathcal{D}_{tr} \cup \mathcal{D}_p, \mathbf{w}). \tag{12}$$

This relaxation enables attack algorithms to optimize the outer problem directly. However, this formulation may be problematic as its training objective does not necessarily reflect its true influence on test data. However, for unlearnable examples, the zero-sum reduction is feasible, and might be the only viable approach. See Appendix B for more details.

This problem is addressed by Koh et al. (2018) with an assumption that the attacker can acquire a *target* model parameter, usually using a label flip attack which considers a much larger poisoning fraction $\varepsilon$. By adding a constraint involving the target parameter $\mathbf{w}_{tar}$, the attacker can search for poisoned points that maximize the loss $\ell$ while keeping a low loss on $\mathbf{w}_*^{tar}$. However, such target parameters are hard to obtain since, as we will demonstrate, non-convex models appear to be robust to label flip attacks and there are no guarantees that $\mathbf{w}_*^{tar}$ is the solution of Equation (7).

(3) Fixed follower (model): Geiping et al. (2020) propose a gradient matching algorithm for crafting targeted poisoning attacks, which can also be easily adapted to *unlearnable examples* (Fowl et al., 2021a). This method fixes the follower and supposes it acquires clean parameter $\mathbf{w}$ on clean data $\mathcal{D}_{tr}$. We define a reversed function $f'$, where $f'$ can be the reversed cross entropy loss for classification problems (Fowl et al., 2021a). As $f'$ discourages the network from classifying clean samples, one can mimic its gradient $\nabla_{\mathbf{w}} f'(\mathbf{w}; \mathcal{D}_{tr})$ by adding poisoned data such that:

$$\nabla_{\mathbf{w}} f'(\mathbf{w}; \mathcal{D}_{tr}) \approx \nabla_{\mathbf{w}} f(\mathbf{w}; \mathcal{D}_{tr} \cup \mathcal{D}_p). \tag{13}$$

To accomplish this goal, Geiping et al. (2020) define a similarity function $\mathcal{S}$ for gradient matching, leading to the attack objective:

$$\mathcal{L} = S(\nabla_{\mathbf{w}} f'(\mathbf{w}; \mathcal{D}_{tr}), \nabla_{\mathbf{w}} f(\mathbf{w}; \mathcal{D}_{tr} \cup \mathcal{D}_p)), \tag{14}$$

where we minimize $\mathcal{L}$ w.r.t $\mathcal{D}_p$. This method is not studied yet in the indiscriminate attack literature, but would serve as an interesting future work.

**TGDA attack.** In this paper, we solve Equation (7) and avoid the calculation of $\frac{\partial \mathbf{w}_*}{\partial \mathcal{D}_p}$ using the Total gradient descent ascent (TGDA) algorithm (Evtushenko, 1974; Fiez et al., 2020) [1]: TGDA takes a total gradient ascent step for the leader and a gradient descent step for the follower:

$$\mathbf{x}_{t+1} = \mathbf{x}_t + \eta_t \mathsf{D}_{\mathbf{x}} \ell(\mathbf{x}_t, \mathbf{w}_t), \tag{15}$$

$$\mathbf{w}_{t+1} = \mathbf{w}_t - \eta_t \nabla_{\mathbf{w}} f(\mathbf{x}_t, \mathbf{w}_t) \tag{16}$$

where $\mathsf{D}_{\mathbf{x}} := \nabla_{\mathbf{x}}\ell - \nabla_{\mathbf{wx}} f \cdot \nabla_{\mathbf{ww}}^{-1} f \cdot \nabla_{\mathbf{w}}\ell$ is the total derivative of $\ell$ with respect to $\mathbf{x}$, which implicitly measures the change of $\mathbf{w}$ with respect to $\mathcal{D}_p$. As optimizing $\ell$ does not involve the attacker parameter $\theta$, we can rewrite $\mathsf{D}_{\mathbf{x}} := -\nabla_{\mathbf{wx}} f \cdot \nabla_{\mathbf{ww}}^{-1} f \cdot \nabla_{\mathbf{w}}\ell$. Here, the product $(\nabla_{\mathbf{ww}}^{-1} f \cdot \nabla_{\mathbf{w}}\ell)$ can be efficiently computed using conjugate gradient (CG) equipped with Hessian-vector products computed by autodiff. As CG is essentially a *Hessian inverse-free approach* (Martens, 2010), each step requires only linear time. Note that TGDA can also be treated as letting $k \to \infty$ in UGDA.

We thus apply the total gradient descent ascent algorithm and call this the **TGDA attack**. Avoiding computing $\frac{\partial \mathbf{w}_*}{\partial \mathcal{D}_p}$ enables us to parameterize $\mathcal{D}_p$ and generate points indirectly by treating $\mathsf{L}$ as a separate model. Namely that $\mathcal{D}_p = \mathsf{L}_\theta(\mathcal{D}'_{tr})$, where $\theta$ is the model parameter and $\mathcal{D}'_{tr}$ is part of the training set to be poisoned. Therefore, we can rewrite Equation (15) as:

$$\theta_{t+1} = \theta_t + \eta_t \mathsf{D}_\theta \ell(\theta_t, \mathbf{w}_t). \tag{17}$$

Thus, we have arrived at a poisoning attack that generates $\mathcal{D}_p$ in a batch rather than individually, which greatly improves the attack efficiency in Algorithm 1. Note that the TGA update does not depend on the choice of $\varepsilon$. This is a significant advantage over previous methods as the running time does not increase as the attacker is allowed a larger budget of introduced poisoned points, thus enabling data poisoning attacks on larger training sets.

---

**Algorithm 1:** TGDA Attack

---

**Input:** Training set $\mathcal{D}_{tr} = \{x_i, y_i\}_{i=1}^N$, validation set $\mathcal{D}_v$, training steps $T$, attacker step size $\alpha$, attacker number of steps $m$, defender step size $\beta$, defender number of steps $n$, poisoning fraction $\varepsilon$, $\mathsf{L}$ with $\theta_{pre}$ and $\ell = \mathcal{L}(\mathcal{D}_v, \mathbf{w}_*)$ , $\mathsf{F}$ with $\mathbf{w}_{pre}$ and $f = \mathcal{L}(\mathcal{D}_{tr} \cup \mathcal{D}_p, \mathbf{w})$.

**1** Initialize poisoned data set $\mathcal{D}_p^0 \longleftarrow \{(x'_1, y'_1), ..., (x'_{\varepsilon N}, y'_{\varepsilon N})\}$

**2 for** $t = 1, ..., T$ **do**

**3**   **for** $i = 1, ..., m$ **do**

**4**     $\lfloor$ $\theta \leftarrow \theta + \alpha \mathsf{D}_\theta \ell(\theta, \mathbf{w}_t)$                                    // TGA on $\mathsf{L}$

**5**   **for** $j = 1, ..., n$ **do**

**6**     $\lfloor$ $\mathbf{w} \leftarrow \mathbf{w} - \beta \nabla_{\mathbf{w}} f(\theta, \mathbf{w})$                                    // GD on $\mathsf{F}$

**7** return model $\mathsf{L}_\theta$ and poisoned set $\mathcal{D}_p = \mathsf{L}_\theta(\mathcal{D}_p^0)$

---

**Necessity of Stackelberg game.** Although Equation (7) is equivalent to the bilevel optimization problem in Muñoz-González et al. (2017); Huang et al. (2020); Koh et al. (2018), our sequential Stackelberg formulation is more suggestive of the data poisoning problem as it reveals the subtlety in the order of the attacker and the defender.

## 4 Implementation

In this section, we (1) discuss the limitations of existing data poisoning attacks and how to address them, (2) set an efficient attack architecture for the TGDA attack.

---

[1]There are other possible solvers for Equation (7), and we have listed them in Appendix C.

Table 1: Summary of existing poisoning attack algorithms, evaluations, and their respective code. While some papers may include experiments on other datasets, we only cover vision datasets as our main focus is image classification. The attacks: Random label flip and Adversarial label flip attacks (Biggio et al., 2011), P-SVM: PoisonSVM attack (Biggio et al., 2011), Min-max attack (Steinhardt et al., 2017), KKT attack (Koh et al., 2018), i-Min-max: improved Min-max attack (Koh et al., 2018), MT: Model Targeted attack (Suya et al., 2021), BG: Back-gradient attack (Muñoz-González et al., 2017).

| Attack | Dataset | Model | $|\mathcal{D}_{tr}|$ | $|\mathcal{D}_{test}|$ | $\varepsilon$ | Code | Multiclass | Batch |
|---|---|---|---|---|---|---|---|---|
| Random label flip | toy | SVM | / | / | 0-40% | git | ✓ | $\varepsilon|\mathcal{D}_{tr}|$ |
| Adversarial label flip | toy | SVM | / | / | 0-40% | git | ✗ | $\varepsilon|\mathcal{D}_{tr}|$ |
| P-SVM | MNIST-17 | SVM | 100 | 500 | 0-9% | git | ✗ | 1 |
| Min-max | MNIST-17/Dogfish | SVM | 60000 | 10000 | 0-30% | git | ✓ | 1 |
| KKT | MNIST-17/Dogfish | SVM, LR | 13007/1800 | 2163/600 | 3% | git | ✗ | 1 |
| i-Min-max | MNIST | SVM | 60000 | 10000 | 3% | git | ✓ | 1 |
| MT | MNIST-17/Dogfish | SVM, LR | 13007/1800 | 2163/600 | / | git | ✓ | 1 |
| BG | MNIST | SVM, NN | 1000 | 8000 | 0-6% | git | ✓ | 1 |

## 4.1 Current Limitations

We observe two limitations of existing data poisoning attacks.

**Limitation 1: Inconsistent assumptions.** We first summarize existing indiscriminate data poisoning attacks in Table 1, where we identify that such attacks work under subtly different assumptions, on, for example, the attacker's knowledge, the attack formulation, and the training set size. These inconsistencies result in somewhat unfair comparisons between methods.

Solution: We set an experimental protocol for generalizing existing attacks and benchmarking data poisoning attacks for systematic analysis in the future. Here we fix three key variants:

(1) the attacker's knowledge: as discussed in Section 2, we consider training-only attacks;

(2) the attack formulation: in Section 3, we introduce three possible formulations, namely non-zero-sum, zero-sum, and zero-sum with target parameters. We will show in the experiment section that the latter two are ineffective against neural networks.

(3) the dataset size: existing works measure attack efficacy with respect to the size of the poisoned dataset, where size is measured as a *fraction* $\varepsilon$ of the training dataset. However, some works subsample and thus reduce the size of the training dataset. As we show in Figure 1, attack efficacy is *not* invariant to the size of the training set: larger training sets appear to be harder to poison. Furthermore, keeping $\varepsilon$ fixed, a smaller training set reduces the number of poisoned data points and thus the time required for methods that generate points sequentially, potentially concealing a prohibitive runtime for poisoning the full training set. Thus we consider not only a fixed $\varepsilon$, but also the complete training set for attacks.

**Limitation 2: Running time**. As discussed in Section 3, many existing attacks approach the problem by optimizing individual points directly, thus having to generate poisoned points one by one. Such implementation takes enormous running time (see Section 5) and does not scale to bigger models or datasets.

Solution: We design a new poisoning scheme that allows simultaneous and coordinated generation of $\mathcal{D}_p$ in batches requiring only one pass. Thanks to the TGDA attack in Section 3, we can treat L as a separate model (typically a neural network such as an autoencoder)

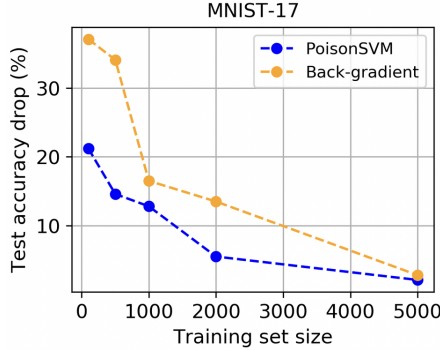

Figure 1: Comparing the efficacy of poisoning MNIST-17 with the PoisonSVM and Back-gradient attacks. The training set size is varied, while the ratio of the number of poisoned points to the training set size is fixed at 3%. These attacks become less effective as training set sizes increase.

that takes part of the $\mathcal{D}_{tr}$ as input and generates $\mathcal{D}_p$ correspondingly. Thus we fix the input and optimize only the parameters of L.

## 4.2   A more efficient attack architecture

Once we have fixed the attack assumptions and poisoned data generation process, we are ready to specify the complete three-stage attack architecture, which enables us to compare poisoning attacks fairly. One can easily apply this unified framework for more advanced attacks in the future.

**(1) Pretrain:** The goals of the attacker L are to: (a) Reduce the test accuracy (i.e., successfully attack). (b) Generate $\mathcal{D}_p$ that is close to $\mathcal{D}_{tr}$ (i.e., thwart potential defenses).

The attacker achieves the first objective during the attack by optimizing $\ell$. However, $\ell$ does not enforce that the distribution of the poisoned points will resemble those of the training set. To this end, we pretrain L to reconstruct $\mathcal{D}_{tr}$, producing a parameter vector $\theta_{pre}$. This process is identical to training an autoencoder.

For the defender, we assume that F is fully trained to convergence. Thus we perform standard training on $\mathcal{D}_{tr}$ to acquire F with $\mathbf{w}_{pre}$. Here we record the performance of F on $\mathcal{D}_{test}$ (denoted as $\texttt{acc}_1$ for image classification tasks) as the benchmark we are poisoning.

**(2) Attack:** We generate poisoned points using the TGDA attack. We assume that the attacker can inject $\varepsilon N$ poisoned points, where $N = |\mathcal{D}_{tr}|$ and $\varepsilon$ is the power of the attacker, measured as a fraction of the training set size. We summarize the attack procedure in Figure 2.

**Initialization:** We take the pretrained model L with parameter $\theta_{pre}$ and F with pretrained parameter $\mathbf{w}_{pre}$ as initialization of the two networks; the complete training set $\mathcal{D}_{tr}$; a validation set $\mathcal{D}_v$ and part of the training set as initialization of the poisoned points $\mathcal{D}_p^0 = \mathcal{D}_{tr}[0 : \varepsilon N]$.

**TGDA attack**: In this paper, we run the TGDA attack to generate poisoned data. But it can be changed to any suitable attack for comparison.

Specifically, we follow Algorithm 1 and perform $m$ steps of TGA updates for the attacker, and $n$ steps of GD updates for the defender in one pass. We discuss the role of $m$ and $n$ in Section 5.

Note that previous works (e.g., Koh et al. 2018; Muñoz-González et al. 2017) choose $n = 1$ by default. However, we argue that this is not necessarily appropriate. When a system is deployed, the model is generally trained until convergence rather than for only a single step. Thus we recommend choosing a much larger $n$ (e.g., $n = 20$ in our experiments) to better resemble the testing scenario.

**Label Information:** We specify that $\mathcal{D}_p^0 = \{x_i, y_i\}_{i=1}^{\varepsilon N}$. Prior works (e.g., Koh et al. 2018; Muñoz-González et al. 2017) optimize $x$ to produce $x_p$, and perform a label flip on $y$ to produce $y_p$ (more details in Appendix A). This approach neglects label information during optimization.

In contrast, we **fix** $y_p = y$, and concatenate $x$ and $y$ to $\mathcal{D}_p^0 = \{x_i; y_i\}_{i=1}^{\varepsilon N}$ as input to L. Thus we generate poisoned points by considering the label information. We emphasize that we do not optimize or change the label during the attack, but merely use it to aid the construction of the poisoned $x_p$. Thus, our attack can be categorized as clean label.

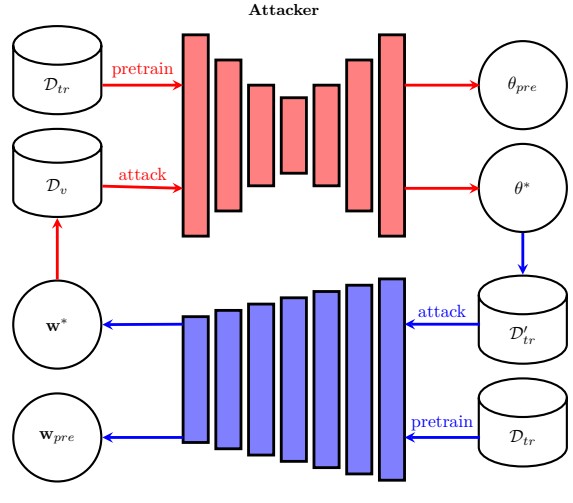

Figure 2: Our experimental protocol benchmarks data poisoning attacks. (1) Pretrain: the attacker and the defender are first trained on $\mathcal{D}_{tr}$ to yield a good autoencoder/classifier respectively. (2) During the attack, the attacker generates the optimal $\theta^*$ (thus $\mathcal{D}_p$) w.r.t $\mathcal{D}_v$ and the the optimal $\mathbf{w}^*$; the defender generates optimal $\mathbf{w}^*$ w.r.t $\mathcal{D}'_{tr} = \mathcal{D}_{tr} \cup \mathcal{D}_p$ and the optimal $\theta^*$ (which mimics testing).

**(3) Testing**: Finally, we discuss how we measure the effectiveness of an attack. In a realistic setting, the testing procedure should be identical to the pretrain procedure, such that we can measure the effectiveness of $\mathcal{D}_p$ fairly. The consistency between pretrain and testing is crucial as the model $\mathsf{F}$ is likely to underfit with fewer training steps.

Given the final $\theta$, we produce the poisoned points $\mathcal{D}_p = \mathsf{L}_\theta(\mathcal{D}_p^0)$ and train $\mathsf{F}$ from scratch on $\mathcal{D}_{tr} \cup \mathcal{D}_p$. Finally, we acquire the performance of $\mathsf{F}$ on $\mathcal{D}_{test}$ (denoted as $\texttt{acc}_2$ for image classification tasks). By comparing the discrepancy between pretrain and testing $\texttt{acc}_1 - \texttt{acc}_2$ we can evaluate the efficacy of an indiscriminate data poisoning attack.

## 5 Experiments

We evaluate our TGDA attack on various models for image classification tasks and show the efficacy of our method for poisoning neural networks. In comparison to existing indiscriminate data poisoning attacks, we show that our attack is superior in terms of both effectiveness and efficiency.

Specifically, our results confirm the following: (1) By applying the Stackelberg game formulation and incorporating second-order information, we can attack neural networks with improved efficiency and efficacy using the TGDA attack. (2) The efficient attack architecture further enables the TGDA attack to generate $\mathcal{D}_p$ in batches. (3) The poisoned points are visually similar to clean data, making the attack intuitively resistant to defenses.

### 5.1 Experimental Settings

**Hardware and package:** Experiments were run on a cluster with `T4` and `P100` GPUs. The platform we use is PyTorch (Paszke et al., 2019). Specifically, autodiff can be easily implemented using `torch.autograd`. As for the total gradient calculation, we follow Zhang et al. (2021) and apply conjugate gradient for calculating Hessian-vector products.

**Dataset:** We consider image classification on MNIST (Deng, 2012) (60,000 training and 10,000 test images), and CIFAR-10 (Krizhevsky, 2009) (50,000 training and 10,000 test images) datasets. We are not aware of prior work that performs indiscriminate data poisoning on a dataset more complex than MNIST or CIFAR-10, and, as we will see, even these settings give rise to significant challenges in designing efficient and effective attacks. Indeed, some prior works consider only a simplified subset of MNIST (e.g., binary classification on 1's and 7's, or subsampling the training set to 1,000 points) or CIFAR-10 (e.g., binary classification on dogs and fish). In contrast, we set a benchmark by using the full datasets for multiclass classification.

**Training and validation set:** During the attack, we need to split the clean training data into the training set $\mathcal{D}_{tr}$ and validation set $\mathcal{D}_v$. Here we split the data to 70% training and 30% validation, respectively. Thus, for the MNIST dataset, we have $|\mathcal{D}_{tr}| = 42000$ and $|\mathcal{D}_v| = 18000$. For the CIFAR-10 dataset, we have $|\mathcal{D}_{tr}| = 35000$ and $|\mathcal{D}_v| = 15000$.

**Attacker models and Defender models:** (1) For the attacker model, for MNIST dataset: we use a three-layer neural network, with three fully connected layers and leaky ReLU activations; for CIFAR-10 dataset, we use an autoencoder with three convoluational layers and three conv transpose layers. The attacker takes the concatenation of the image and the label as the input, and generates the poisoned points. (2) For the defender, we examine three target models for MNIST: Logistic Regression, a neural network (NN) with three layers and a convolutional neural network (CNN) with two convolutional layers, maxpooling and one fully connected layer; and only the CNN model and ResNet-18 (He et al., 2016) for CIFAR-10 (as CIFAR-10 contains RBG images).

**Hyperparameters:** (1) Pretrain: we use a batch size of 1,000 for MNIST and 256 for CIFAR-10, and optimize the network using our own implementation of gradient descent with `torch.autograd`. We choose the learning rate as 0.1 and train for 100 epochs. (2) Attack: for the attacker, we choose $\alpha = 0.01$, $m = 1$ by default; for the defender, we choose $\beta = 0.1$, $n = 20$ by default. We set the batch size to be 1,000 for MNIST; 256 for CIFAR10 and train for 200 epochs, where the attacker is updated using total gradient ascent and the defender is updated using gradient descent. We follow Zhang et al. (2021) and implement TGA using

Table 2: The attack accuracy/accuracy drop (%) and attack running time (hours) on the MNIST dataset. We only record the attack running time since pretrain and testing time are fixed across different methods. As the label flip attack does not involve optimization, its running time is always 0. We take three different runs for TGDA to get the mean and the standard derivation. Our attack outperforms the Min-max, i-Min-max and Back-gradient attacks in terms of both effectiveness and efficiency across neural networks.

| Model | Clean | Label Flip | | Min-max | | i-Min-max | | BG | | TGDA(ours) | |
|---|---|---|---|---|---|---|---|---|---|---|---|
| | Acc | Acc/Drop | Time | Acc/Drop | Time | Acc/Drop | Time | Acc/Drop | Time | Acc/Drop | Time |
| LR | 92.35 | 90.83 / 1.52 | 0 hrs | 89.80/2.55 | 0.7 hrs | 89.56/**2.79** | 19 hrs | 89.82 / 2.53 | 27 hrs | 89.56 / **2.79**$_{\pm 0.07}$ | 1.1 hrs |
| NN | 98.04 | 97.99 / 0.05 | 0 hrs | 98.07/-0.03 | 13 hrs | 97.82/0.22 | 73 hrs | 97.67 / 0.37 | 239 hrs | 96.54 / **1.50**$_{\pm 0.02}$ | 15 hrs |
| CNN | 99.13 | 99.12 / 0.01 | 0 hrs | 99.55/-0.42 | 63hrs | 99.05/0.06 | 246 hrs | 99.02 / 0.09 | 2153 hrs | 98.02 / **1.11**$_{\pm 0.01}$ | 75 hrs |

conjugate gradient. We choose the poisoning fraction $\varepsilon = 3\%$ by default. Note that choosing a bigger $\varepsilon$ will not increase our running time, but we choose a small $\varepsilon$ to resemble the realistic setting in which the attacker is limited in their access to the training data. (3) Testing: we choose the exact same setting as pretrain to keep the defender's training scheme consistent.

**Baselines:** There is a spectrum of data poisoning attacks in the literature. However, due to their attack formulations, only a few attacks can be directly compared with our method. See Table 1 in Appendix A for a complete summary. For instance, the Poison SVM (Biggio et al., 2011) and KKT (Koh et al., 2018) attacks can only be applied to convex models for binary classification; the Min-max (Steinhardt et al., 2017) and the Model targeted (Suya et al., 2021) attacks can be only applied to convex models. However, it is possible to modify Min-max (Steinhardt et al., 2017) and i-Min-max (Koh et al., 2018) attacks to attack neural networks. Moreover, we compare with two baseline methods that can originally attack neural networks: the Back-gradient attack (Muñoz-González et al., 2017) and the Label flip attack (Biggio et al., 2011). It is also possible to apply certain targeted attack methods (e.g., MetaPoison, Huang et al. 2020) in the context of indiscriminate attacks. Thus we compare with MetaPoison on CIFAR-10 under our unified architecture. We follow Huang et al. (2020) and choose $K = 2$ unrolled inner steps, 60 outer steps, and an ensemble of 24 inner models.

## 5.2 Comparison with Benchmarks

**MNIST.** We compare our attack with the Min-max, i-Min-max, Back-gradient and the Label flip attacks with $\varepsilon = 3\%$ on MNIST in Table 2. Since the Min-max, i-Min-max, and Back-gradient attack relies on generating poisoned points sequentially, we cannot adapt it into our unified architecture and run their code directly for comparison. For the label flip attack, we flip the label according to the rule $y \leftarrow 10 - y$, as there are 10 classes in MNIST.

We observe that label flip attack, though very efficient, is not effective against neural networks. Min-max attack, due to its zero-sum formulation, does not work on neural networks. i-Min-max attack is effective against LR, but performs poorly on neural networks where the assumption of convexity fails. Although Muñoz-González et al. (2017) show empirically that the Back-gradient attack is effective when attacking subsets of MNIST (1,000 training samples, 5,000 testing samples), we show that the attack is much less effective on the full dataset. We also observe that the complexity of the target model affects the attack effectiveness significantly. Specifically, we find that neural networks are generally more robust against indiscriminate data poisoning attacks, among which, the CNN architecture is even more robust. Overall, our method outperforms the baseline methods across the three target models. Moreover, with our unified architecture, we significantly reduce the running time of poisoning attacks.

**CIFAR-10.** We compare our attack with the Label flip attack and the MetaPoison attack with $\varepsilon = 3\%$ on CIFAR-10 in Table 3. We omit comparison with the Back-gradient attack as it is too computationally expensive to run on CIFAR-10. We observe that running the TGDA attack following Algorithm 1 directly is computationally expensive on large models (e.g., ResNet, He et al. 2016). However, it is possible to run TGDA on such models by slightly changing Algorithm 1: we split the dataset into 8 partitions and run TGDA separately on different GPUs. This simple trick enables us to poison deeper models and we find it works well in practice. We observe that the TGDA attack is very effective at poisoning the CNN and the ResNet-18 architectures, Also, MetaPoison is a more efficient attack (meta-learning with two unrolled steps is

Table 3: The attack accuracy/accuracy drop (%) and attack running time (hours) on CIFAR-10. Note that TGDA experiments are performed on 8 GPUs for parallel training. We take three different runs for TGDA and MetaPoison to get the mean and the standard derivation.

| Model | Clean | Label Flip | | MetaPoison | | TGDA(ours) | |
|---|---|---|---|---|---|---|---|
| | Acc | Acc/Drop | Time | Acc/Drop | Time | Acc/Drop | Time |
| CNN | 69.44 | 68.99/0.45 | 0 hrs | $68.14/1.13_{\pm 0.12}$ | 35 hrs | $65.15/4.29_{\pm 0.09}$ | 42 hrs |
| ResNet-18 | 94.95 | 94.79/0.16 | 0 hrs | $92.90/2.05_{\pm 0.07}$ | 108 hrs | $89.41/5.54_{\pm 0.03}$ | 162 hrs |

Table 4: Comparing the TGDA attack with different orders: attacker as the leader and defender as the leader in terms of test accuracy/accuracy drop(%). Attacks are more effective when the attacker is the leader.

| Target Model | Clean | Attacker as leader | Defender as leader |
|---|---|---|---|
| LR | 92.35 | 89.56 / **2.79** | 89.79 / 2.56 |
| NN | 98.04 | 96.54 / **1.50** | 96.98 / 1.06 |
| CNN | 99.13 | 98.02 / **1.11** | 98.66 / 0.47 |

much quicker than calculating total gradient), but since its original objective is to perform targeted attacks, its application to indiscriminate attacks is not effective. Moreover, the difference between the efficacy of the TGDA attack on MNIST and CIFAR-10 suggests that indiscriminate attacks may be dataset dependent, with MNIST being harder to poison than CIFAR-10.

## 5.3 Ablation Studies

To better understand our TGDA attack, we perform ablation studies on the order in the Stackelberg game, the attack formulation, roles in our unified attack framework, and the choice of hyperparameters. For computational considerations, we run all ablation studies on the MNIST dataset unless specified. Furthermore, we include empirically comparison with *unlearnable examples* in Appendix B.

**Who acts first.** In Section 3, we assume that the attacker is the leader and the defender is the follower, i.e., that the attacker acts first. Here, we examine the outcome of reversing the order, where the defender acts first. Table 4 shows the comparison. We observe that across all models, reversing the order would result in a less effective attack. This result shows that even without any defense strategy, the target model would be more robust if the defender acts one step ahead of the attacker.

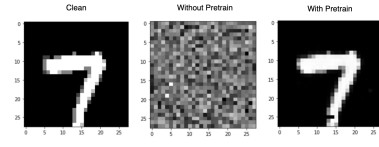

Figure 3: We visualize the poisoned data generated by the TGDA attack with/without pretraining the leader L on the MNIST dataset.

**Attack formulation.** In Section 3, we discuss a relaxed attack formulation, where $\ell = f$ and the game is zero-sum. We perform experiments on this setting and show results in Table 5. We observe that the non-zero-sum formulation is significantly more effective, and in some cases, the zero-sum setting actually *increases* the accuracy after poisoning. We also find that using target parameters would not work for neural networks as they are robust to label flip attacks even when $\varepsilon$ is large. We ran a label flip attack with $\varepsilon = 100\%$ and observed only 0.1% and 0.07% accuracy drop on NN and CNN architectures, respectively. This provides further evidence that neural networks are robust to massive label noise, as previously observed by Rolnick et al. (2017).

**Role of pretraining.** In Section 4, we propose two desired properties of L, among which L should generate $\mathcal{D}_p$ that is visually similar to $\mathcal{D}_{tr}$. Thus requires the pretraining of L for reconstructing images. We perform experiments without pretraining L to examine its role in effecting the attacker. Figure 3 confirms that without pretraining, the attacker will generate images that are visually different from the $\mathcal{D}_{tr}$ distribution, thus fragile to possible defenses. Moreover, Table 6 indicates that without pretraining L, the attack will also be ineffective. Thus we have demonstrated the necessity of the visual similarity between $\mathcal{D}_p$ and $\mathcal{D}_{tr}$.

**Different $\varepsilon$.** We have set $\varepsilon = 3\%$ in previous experiments. However, unlike prior methods which generate points one at a time, the running time of our attack does not scale with $\varepsilon$, and thus we can consider significantly larger $\varepsilon$ and compare with other feasible methods. Figure 4 shows that attack efficacy increases

Table 5: Comparing the TGDA attack with different formulations: non-zero-sum and zero-sum in terms of test accuracy/accuracy drop (%). Non-zero-sum is more effective at generating poisoning attacks.

| Target Model | Clean | Non Zero-sum | Zero-sum |
|:---:|:---:|:---:|:---:|
| LR | 92.35 | 89.56 / **2.79** | 92.33 / 0.02 |
| NN | 98.04 | 96.54 / **1.50** | 98.07 / -0.03 |
| CNN | 99.13 | 98.02 / **1.11** | 99.55 / -0.42 |

Table 6: Comparing the TGDA attack with/without pretraining the attacker L in terms of test accuracy/accuracy drop (%). Pretraining strongly improves attack efficacy.

| Target Model | Clean | With Pretrain | Without Pretrain |
|:---:|:---:|:---:|:---:|
| LR | 92.35 | 89.56 / **2.79** | 92.09 / 0.26 |
| NN | 98.04 | 96.54 / **1.50** | 97.47 / 0.57 |
| CNN | 99.13 | 98.02 / **1.11** | 98.72 / 0.41 |

with $\varepsilon$, but the accuracy drop is significantly less than $\varepsilon$ when $\varepsilon$ is very large. Moreover, TGDA outperforms baseline methods across any choice of $\varepsilon$.

**Number of steps $m$ and $n$.** We discuss the choice of $m$ and $n$, the number of steps of L and F, respectively. We perform three choices of $m$ and $n$ in Table 7. We observe that 20 steps of TGA and 1 step of GD results in the most effective attack. This indicates that when $m > n$, the outer maximization problem is better solved with more TGA updates. However, setting 4 ($m = 20, n = 1$) takes 10 times more computation than setting 3 ($m = 1, n = 20$), due to the fact that the TGA update is expensive. When $m = n = 1$, the attack is less effective as the defender might not be fully trained to respond to the attack. When $n = 0$, the attack is hardly effective at all as the target model is not retrained. We conclude that different choices of $m$ and $n$ would result in a trade-off between effectiveness and efficiency.

**Cold-Start.** The methods we compare in this work all belong to the partial warm-start category for bilevel optimization (Vicol et al., 2022). It is also possible to formulate the cold-start Stackelberg game for data poisoning. Specifically, we follow Vicol et al. (2022) such that in Algorithm 1, the follower update is modified to $\mathbf{w} \leftarrow \mathbf{w}_{pre} - \beta \nabla_\mathbf{w} f(\theta, \mathbf{w})$. We report the results in Table 8 on MNIST dataset. We observe that in indiscriminate data poisoning, partial warm-start is a better approach than cold-start overall, and our outer problem (autoencoder for generating poisoned points) does not appear to be highly over-parameterized.

### 5.4 Visualization of attacks

Finally, we visualize some poisoned points $\mathcal{D}_p$ generated by the TGDA attack in Figure 5.

The poisoning samples against NN and CNN are visually very similar with $\mathcal{D}_{tr}$, as our attack is a clean label attack (see Section 4). Moreover, we evaluate the magnitude of perturbation by calculating the maximum of pixel-level difference. Both visual similarity and magnitude of perturbation provide heuristic evidence that the TGDA attack may be robust against data sanitization algorithms. Note that $\mathcal{D}_p$ against LR is visually distinguishable, and the reason behind this discrepancy between the convex model and the neural networks

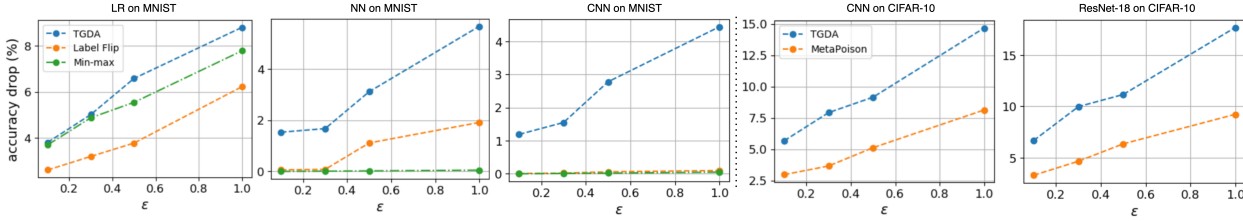

Figure 4: Accuracy drop induced by our TGDA poisoning attack and baseline methods versus $\varepsilon$ (left three figures: MNIST; right two figures: CIFAR-10). Attack efficacy increases modestly with $\varepsilon$. Note that when $\varepsilon = 1$, only 50% of the training set is filled with poisoned data.

Table 7: Comparing different numbers of steps of the attacker ($m$) and defender ($n$) in terms of test accuracy/accuracy drop (%). Many attacker steps and a single defender step produces the most effective attacks.

| Model | Clean | $m = 1, n = 0$ | $m = 1, n = 1$ | $m = 1, n = 20$ | $m = 20, n = 1$ | $m = n = 20$ |
|---|---|---|---|---|---|---|
| LR | 92.35 | 92.30 / 0.05 | 89.97 / 2.38 | 89.56 / 2.79 | 89.29 / **3.06** | 89.77 / 2.57 |
| NN | 98.04 | 98.02 / 0.02 | 97.03 / 1.01 | 96.54 / 1.50 | 96.33 / **1.71** | 96.85 / 1.19 |

Table 8: Comparing the TGDA attack with partial warm-start (our original setting) and cold-start in terms of test accuracy/accuracy drop (%). Cold-start training is less effective overall.

| Model | Clean | Partial Warm-start | Cold-start |
|---|---|---|---|
| LR | 92.35 | 89.56 / **2.79** | 89.84 / 2.41 |
| NN | 98.04 | 96.54 / **1.50** | 96.77 / 1.27 |
| CNN | 99.13 | 98.02 / **1.11** | 98.33 / 0.80 |

may be that the attacker L is not expressive enough to generate extremely strong poisoning attacks against neural networks.

## 5.5 Transferability of the TGDA attack

Even for training-only attacks, the assumption on the attacker's knowledge can be too strong. Thus we study the scenario when the attacker has limited knowledge regarding the defender's model F and training process, where the attacker has to use a surrogate model to simulate the defender. We report the transferability of the TGDA attack on different surrogate models in Table 9. We observe that poisoned points generated against LR and NN have a much lower impact against other models, while applying CNNs as the surrogate model is effective on all models.

## 5.6 Against Defenses:

To further evaluate the robustness of the TGDA attack against data sanitization algorithms:

(a) We perform the loss defense (Koh et al., 2018) by removing 3% of training points with the largest loss. We compare with pGAN (Muñoz-González et al., 2019), which includes a constraint on the similarity between the clean and poisoned samples, and is thus inherently robust against defenses. In Table 10, we observe that although we do not add an explicit constraint on detectability in our loss function, our method still reaches comparable robustness against such defenses with pGAN.

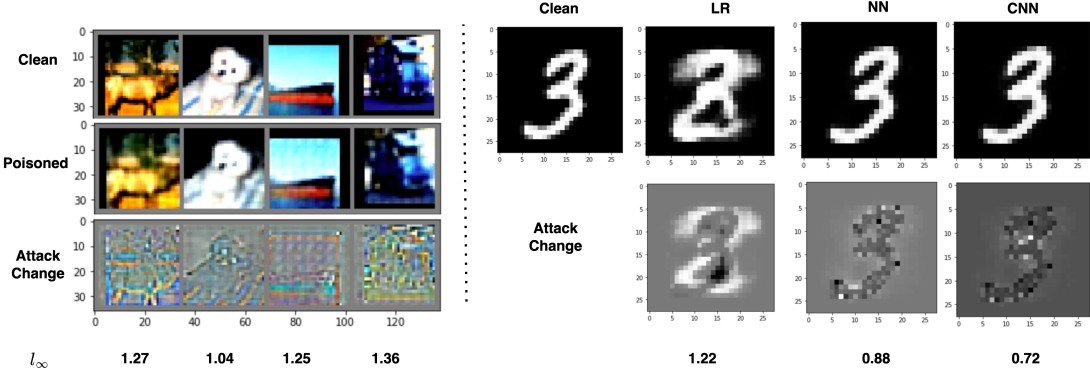

Figure 5: We visualize the poisoned data generated by the TGDA attack and report the magnitude of perturbation (left: CIFAR-10; right: MNIST).

Table 9: Transferability expeirments on MNIST.

| Surrogate | **LR** | | | **NN** | | | **CNN** | | |
|---|---|---|---|---|---|---|---|---|---|
| Target | LR | NN | CNN | LR | NN | CNN | LR | NN | CNN |
| Accuracy Drop(%) | 2.79 | 0.12 | 0.27 | 0.13 | 1.50 | 0.62 | 3.22 | 1.47 | 1.11 |

Table 10: Comparison with pGAN on MNIST with loss defense.

| Method | **TGDA (wo/w defense)** | | | **pGAN(wo/w defense)** | | |
|---|---|---|---|---|---|---|
| Target Model | LR | NN | CNN | LR | NN | CNN |
| Accuracy Drop (%) | 2.79/2.56 | 1.50/1.49 | 1.11/1.104 | 2.52/2.49 | 1.09/1.07 | 0.74/0.73 |

Table 11: TGDA attack on MNIST with Influence/Sever/MaxUp defense.

| **Model** | Influence | | Sever | | MaxUp | |
|---|---|---|---|---|---|---|
| | wo defense | w defense | wo defense | w defense | wo defense | w defense |
| LR | 2.79 | 2.45 | 2.79 | 2.13 | 2.79 | 2.77 |
| NN | 1.50 | 1.48 | 1.50 | 1.32 | 1.50 | 1.50 |
| CNN | 1.11 | 1.10 | 1.11 | 0.98 | 1.11 | 1.11 |

(b) Other defenses remove suspicious points according to their effect on the learned parameters, e.g., through influence functions (influence defense in Koh & Liang 2017) or gradients (Sever in Diakonikolas et al. 2016). Specifically, influence defense removes 3% of training points with the highest influence, defined using their gradients and Hessian-vector products (Koh & Liang, 2017); Sever removes 3% of training points with the highest outlier scores, defined using the top singular value in the matrix of gradients. Here we examine the robustness of TGDA against these two strong defenses. We observe in Table 11 that TGDA is robust against Influence defense, but its effectiveness is significantly reduced by Sever. Thus, we conclude Sever is potentially a good defense against the TGDA attack, and it might require special design (e.g., an explicit constraint on the singular value) to break Sever sanitation.

(c) We examine the robustness of our TGDA attack against strong data augmentations, e.g., the MaxUp defense[2] of Gong et al. (2020). In a nutshell, MaxUp generates a set of augmented data with random perturbations and then aims at minimizing the worst case loss over the augmented data. This training technique addresses overfitting and serves as a possible defense against adversarial examples. However, it is not clear if MaxUp is a good defense against indiscriminate data poisoning attacks. Thus, we implement MaxUp under our testing protocol, where we add random perturbations to the training and the poisoned data, i.e., $\{\mathcal{D}_{tr} \cup \mathcal{D}_p\}$, and then minimize the worst case loss over the augmented set. We report the results in Table 11, where we observe that even though MaxUp is a good defense against adversarial examples, it is not readily an effective defense against indiscriminate data poisoning attacks. Part of the reason we believe is that in our formulation the attacker anticipates the retraining done by the defender, in contrast to the adversarial example setting.

## 6 Conclusions

While indiscriminate data poisoning attacks have been well studied under various formulations and settings on convex models, non-convex models remain significantly underexplored. Our work serves as a first exploration into poisoning neural networks under a unified architecture. While prior state-of-the-art attacks failed at this task due to either the attack formulation or a computationally prohibitive algorithm, we propose a novel Total Gradient Descent Ascent (TGDA) attack by exploiting second-order information, which enables generating thousands of poisoned points in only one pass. We perform experiments on (convolutional) neural networks and empirically demonstrate the feasibility of poisoning them. Moreover, the TGDA attack

---

[2]We follow the implementation in https://github.com/Yunodo/maxup

produces poisoned samples that are visually indistinguishable from unpoisoned data (i.e., it is a clean-label attack), which is desired in the presence of a curator who may attempt to sanitize the dataset.

Our work has some limitations. While our algorithm is significantly faster than prior methods, it remains computationally expensive to poison deeper models such as ResNet, or larger datasets such as ImageNet. Similarly, while our attacks are significantly more effective than prior methods, we would ideally like a poison fraction of $\varepsilon$ to induce an accuracy drop far larger than $\varepsilon$, as appears to be possible for simpler settings (Lai et al., 2016; Diakonikolas et al., 2016; 2019). We believe our work will set an effective benchmark for future work on poisoning neural networks.

## Acknowledgement

We thank the action editor and reviewers for the constructive comments and additional references, which have greatly improved our presentation and discussion.

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

# A    Indiscriminte data poisoning attacks

We first show that perfect knowledge attacks and training-only attacks can be executed by solving a non-zero-sum bi-level optimization problem.

## A.1    Non-zero-sum setting

For perfect knowledge and training-only attacks, recall that we aim at the following bi-level optimization problem:

$$\max_{\mathcal{D}_p}\ \mathcal{L}(\mathcal{D}_v, \mathbf{w}_*),\ \ \text{s.t.}\ \ \mathbf{w}_* \in \arg\min_{\mathbf{w}\in\mathbb{W}}\ \mathcal{L}(\mathcal{D}_{tr}\cup\mathcal{D}_p, \mathbf{w}), \tag{18}$$

where we constrain $|\mathcal{D}_p| = \varepsilon|\mathcal{D}_{tr}|$ to limit the amount of poisoned data the attacker can inject. The attacker can solve (18) in the training only attack setting. With a stronger assumption where $\mathcal{D}_{test}$ is available, we substitute $\mathcal{D}_v$ with $\mathcal{D}_{test}$ and arrive at the perfect knowledge attack setting.

Existing attacks generate poisoned points one by one by considering the problem:

$$\max_{x_p}\ \mathcal{L}(\mathcal{D}_v, \mathbf{w}_*),\ \ \text{s.t.}\ \ \mathbf{w}_* \in \arg\min_{\mathbf{w}\in\mathbb{W}}\ \mathcal{L}(\mathcal{D}_{tr}\cup\{x_p, y_p\}, \mathbf{w}). \tag{19}$$

While the inner minimization problem can be solved via gradient descent, the outer maximization problem is non-trivial as the dependency of $\mathcal{L}(\mathcal{D}_v, \mathbf{w}_*)$ on $x_p$ is indirectly encoded through the parameter $\mathbf{w}$ of the poisoned model. As a result, we rewrite the desired derivative using the chain rule:

$$\frac{\partial \mathcal{D}(\mathcal{D}_v, \mathbf{w}_*)}{\partial x_p} = \frac{\partial \mathcal{D}(\mathcal{D}_v, \mathbf{w}_*)}{\partial \mathbf{w}_*}\frac{\partial \mathbf{w}_*}{\partial x_p}, \tag{20}$$

where the difficulty lies in computing $\frac{\partial \mathbf{w}_*}{\partial x_p}$, i.e., measuring how much the model parameter $\mathbf{w}$ changes with respect to the poisoning point $x_p$. Various approaches compute $\frac{\partial \mathbf{w}_*}{\partial x_p}$ by solving this problem exactly, using either influence functions (Koh & Liang, 2017) (Influence attack) or KKT conditions (Biggio et al., 2011) (PoisonSVM attack[3]). Another solution is to approximate the problem using gradient descent (Muñoz-González et al., 2017). We discuss each of these approaches below.

**Influence attack.**    The influence function (Hampel, 1974) tells us how the model parameters change as we modify a training point by an infinitesimal amount. Borrowing the presentation from Koh & Liang (2017), we compute the desired derivative as:

$$\frac{\partial \mathbf{w}_*}{\partial x_p} = -H_{\mathbf{w}_*}^{-1}\frac{\partial^2 \mathcal{L}(\{x_p, y_p\}, \mathbf{w}_*)}{\partial \mathbf{w}_* \partial x_p}, \tag{21}$$

where $H_{\mathbf{w}_*}$ is the Hessian of the training loss at $\mathbf{w}_*$:

$$H_{\mathbf{w}_*} := \lambda I + \frac{1}{|\mathcal{D}_{tr}\cup\mathcal{D}_p|}\sum_{(x,y)\in\mathcal{D}_{tr}\cup\mathcal{D}_p}\frac{\partial^2 \mathcal{L}((x,y), \mathbf{w}_*)}{\partial(\mathbf{w}_*)^2} \tag{22}$$

Influence functions are well-defined for convex models like SVMs and are generally accurate for our settings. However, they have been showed to be inaccurate for neural networks (Basu et al., 2021).

**PoisonSVM attack.**    Biggio et al. (2012) replaces the inner problem with its stationary (KKT) conditions. According to the KKT condition, we write the implicit function:

$$\frac{\partial \mathcal{L}(\mathcal{D}_{tr}\cup\{x_p, y_p\}, \mathbf{w}_*)}{\partial \mathbf{w}_*} = 0, \tag{23}$$

---

[3]While this might naturally suggest the name "KKT attack," this name is reserved for a different attack covered in Section A.3.

which yields the linear system:

$$\frac{\partial^2 \mathcal{L}(\mathcal{D}_{tr} \cup \{x_p, y_p\}, \mathbf{w}_*)}{\partial \mathbf{w}_* \partial x_p} + \frac{\partial^2 \mathcal{L}(\mathcal{D}_{tr} \cup \{x_p, y_p\}, \mathbf{w}_*)}{\partial (\mathbf{w}_*)^2} \frac{\partial \mathbf{w}_*}{\partial x_p} = 0, \tag{24}$$

and thus we can solve the desired derivative as:

$$\frac{\partial \mathbf{w}_*}{\partial x_p} = - \left( \frac{\partial^2 \mathcal{L}(\mathcal{D}_{tr} \cup \{x_p, y_p\}, \mathbf{w}_*)}{\partial (\mathbf{w}_*)^2} \right)^{-1} \frac{\partial^2 \mathcal{L}(\mathcal{D}_{tr} \cup \{x_p, y_p\}, \mathbf{w}_*)}{\partial \mathbf{w}_* \partial x_p}. \tag{25}$$

Note that despite their differences in approaching the derivative, both the influence attack and PoisonSVM attack involve the inverse Hessian.

**Back-gradient attack.** Muñoz-González et al. (2017) avoid solving the outer maximization problem exactly by replacing it with a set of iterations performed by an optimization method such as gradient descent. This incomplete optimization of the inner problem allows the algorithm to run faster than the two above methods, and poisoning neural networks.

## A.2 Zero-sum Setting

Steinhardt et al. (2017) reduce Equation (18) to a zero-sum game by replacing $\mathcal{L}(\mathcal{D}_v, \mathbf{w}_*)$ with $\mathcal{L}(\mathcal{D}_{tr} \cup \mathcal{D}_p, \mathbf{w}_*)$, and the original problem can be written as:

$$\max_{\mathcal{D}_p} \ \mathcal{L}(\mathcal{D}_{tr} \cup \mathcal{D}_p, \mathbf{w}_*), \quad \text{s.t.} \quad \mathbf{w}_* \in \arg\min_{\mathbf{w} \in \mathbb{W}} \ \mathcal{L}(\mathcal{D}_{tr} \cup \mathcal{D}_p, \mathbf{w}). \tag{26}$$

which is identical to the saddle-point or zero-sum problem:

$$\max_{\mathcal{D}_p} \min_{\mathbf{w}} \ \mathcal{L}(\mathcal{D}_{tr} \cup \mathcal{D}_p, \mathbf{w}) \tag{27}$$

For an SVM model, given that the loss function is convex, we can solve (27) by swapping the min and max and expand the problem to:

$$\min_{\mathbf{w}} \ \sum_{(x,y) \in \mathcal{D}_{tr}} \mathcal{L}(\{x, y\}, \mathbf{w}) + \max_{\{x_p, y_p\}} \mathcal{L}(\{x_p, y_p\}, \mathbf{w}), \tag{28}$$

However, we emphasize that this relaxed gradient-based attack is problematic and could be ineffective since the loss on the clean data $\mathcal{D}_{tr}$ could still be low. In other words, the inner maximization does not address the true objective where we want to change the model parameter to cause wrong predictions on clean data. This can be addressed by keeping the loss on the poisoned data small, but this contradicts the problem formulation. One solution to this is to use target parameters in Section A.3.

## A.3 Zero-sum Setting with Target parameters

Gradient-based attacks solve a difficult optimization problem in which the poisoned data $\mathcal{D}_p$ affects the objective through the model parameter $\mathbf{w}_*$. As a result, evaluating the gradient usually involves computing a Hessian, a computationally expensive operation which can not be done in many realistic settings. Koh et al. (2018) propose that if we have a target parameter $\mathbf{w}_*^{tar}$ which maximizes the loss on the test data $\mathcal{L}(\mathcal{D}_{test}, \mathbf{w}_*)$, then the problem simplifies to:

$$\text{find } \mathcal{D}_p, \quad \text{s.t.} \quad \mathbf{w}_*^{tar} = \arg\min_{\mathbf{w} \in \mathbb{W}} \ \mathcal{L}(\mathcal{D}_{tr} \cup \mathcal{D}_p, \mathbf{w}), \tag{29}$$

**KKT attack.** Since the target parameter $\mathbf{w}_*^{tar}$ is pre-specified, the condition can be rewritten as:

$$\mathbf{w}_*^{tar} = \arg\min_{\mathbf{w} \in \mathbb{W}} \ \mathcal{L}(\mathcal{D}_{tr} \cup \mathcal{D}_p, \mathbf{w}) \tag{30}$$

$$= \arg\min_{\mathbf{w} \in \mathbb{W}} \ \sum_{\{x,y\} \in \mathcal{D}_{tr}} \mathcal{L}(\{x, y\}, \mathbf{w}) + \sum_{\{x_p, y_p\} \in \mathcal{D}_p} \mathcal{L}(\{x_p, y_p\}, \mathbf{w}), \tag{31}$$

Again we can use the KKT optimality condition to solve the argmin problem for convex losses:

$$\sum_{\{x,y\}\in\mathcal{D}_{tr}}\mathcal{L}(\{x,y\},\mathbf{w}_*^{tar}) + \sum_{\{x_p,y_p\}\in\mathcal{D}_p}\mathcal{L}(\{x_p,y_p\},\mathbf{w}_*^{tar}) = 0 \tag{32}$$

Thus we can rewrite the problem as:

$$\text{find } \mathcal{D}_p, \quad \text{s.t.} \quad \sum_{\{x,y\}\in\mathcal{D}_{tr}}\mathcal{L}(\{x,y\},\mathbf{w}_*^{tar}) + \sum_{\{x_p,y_p\}\in\mathcal{D}_p}\mathcal{L}(\{x_p,y_p\},\mathbf{w}_*^{tar}) = 0. \tag{33}$$

If this problem has a solution, we can find it by solving the equivalent norm-minimization problem:

$$\min_{\mathcal{D}_p} \left\| \sum_{\{x,y\}\in\mathcal{D}_{tr}}\mathcal{L}(\{x,y\},\mathbf{w}_*^{tar}) + \sum_{\{x_p,y_p\}\in\mathcal{D}_p}\mathcal{L}(\{x_p,y_p\},\mathbf{w}_*^{tar}) \right\|_2^2, \tag{34}$$

where the problem can only be minimized if the KKT condition is satisfied. This attack is called the KKT attack.

Of course, the success of this attack relies on the target parameter $\mathbf{w}_*^{tar}$. Koh et al. (2018) propose to use the label flip attack for such purpose where we use the trained parameter as the target. This attack achieves comparable results to other attacks while being much faster since it can be solved efficiently using grid search for binary classification. Note that for multi-class classification, this algorithm quickly become infeasible.

**Improved min-max.** Koh et al. (2018) applies the target parameters to address the issue for the relaxed gradient-based attack, where we add the following constraint during training:

$$\mathcal{L}(\{x,y\},\mathbf{w}_*^{tar}) \leq \tau, \tag{35}$$

where $\tau$ is a fixed threshold. Thus the attacker can search for poisoned points that maximize loss under the current parameter $\mathbf{w}$ while keeping low loss on the target parameter $\mathbf{w}_*^{tar}$.

**Model Targeted Poisoning.** Suya et al. (2021) propose another algorithm for generating poisoned points using target parameters. This attack considers a different attack strategy from the others, where the attacker adopts an online learning procedure. In this case, the attacker does not have a poison fraction $\varepsilon$ to generate a specific amount of poisoned data. Instead, the attacker aims at reaching a stopping criteria (can be either a desired accuracy drop or desired distance to the target parameter). However, such attacking procedure may cause the poison fraction $\varepsilon$ to be large and it is hard to measure the success of the attack. Thus, we use the same setting as others for fair comparison.

### A.4 Training-data-only attack

In the training-data-only attack setting, since the attacker does not have access to the training procedure, the bi-level optimization methods are not applicable. The remaining strategies focus either on modifying the labels only (i.e., label flip attacks).

**Random label flip attack.** Random label flipping is a very simple attack, which constructs a set of poisoned points by randomly selecting training points and flipping their labels:

$$\mathcal{D}_p = \{(\mathbf{x}_i, \bar{y}_i) : (\mathbf{x}_i, y_i) \in \mathcal{D}_{tr}\} \quad \text{s.t.} \quad |\mathcal{D}_p| = \varepsilon |\mathcal{D}_{tr}|, \tag{36}$$

where for each class $j = 1, \ldots, c$, we set

$$\bar{y}_i = j \text{ with probability } p_j. \tag{37}$$

Note that the weights $\{p_j\}$ may depend on the true label $y_i$. For instance, for binary classification (i.e., $c = 2$), we may set $p_{c+1-y_i} = 1$ in which case $\bar{y}_i$ simply flips the true label $y_i$.

**Adversarial label flip attack.** Biggio et al. (2011) consider an adversarial variant of the label flip attack, where the choice of the poisoned points is not random. This attack requires access to the model and training procedure, and thus is not a training-data-only attack. Biggio et al. (2011) design an attack focused on SVMs. They choose to poison non-support vectors, as these are likely to become support vectors when an SVM is trained on the dataset including these points with flipped labels.

**Label flip for multi-class classification** For binary classification, label flip is trivial. Koh et al. (2018) provides a solution for multi-class classification problem. For marginal based models (for example, SVM), we can write the multi-class hinge loss, where we have (Dogan et al., 2016):

$$\mathcal{L}(\mathbf{w}, (x_i, y_i)) = \max\{0, 1 + \max_{j \neq y_i} \mathbf{w}_j x_i - \mathbf{w}_{y_i} x_i\}, \tag{38}$$

where the choice of $j$ is obvious: we choose the index with the highest function score except the target class $y_i$. Naturally, we can use this index $j$ as the optimal label flip. As for non-convex models, the choice of optimal label flip is not clear. In this case, one can use a heuristic by choosing the class with the biggest training loss.

## B Unlearnable Examples

### B.1 Stackelberg Game on Unlearnable Examples

We recall the general Stackelberg game in Section 3, where the follower F chooses $\mathbf{w}$ to best respond to the action $\mathbf{x}$ of the leader L, through minimizing its loss function $f$:

$$\forall \mathbf{x} \in \mathbb{X} \subseteq \mathbb{R}^d, \quad \mathbf{w}_*(\mathbf{x}) \in \arg\min_{\mathbf{w} \in \mathbb{W}} f(\mathbf{x}, \mathbf{w}), \tag{39}$$

and the leader L chooses $\mathbf{x}$ to maximize its loss function $\ell$:

$$\mathbf{x}_* \in \arg\max_{\mathbf{x} \in \mathbb{X}} \ell(\mathbf{x}, \mathbf{w}_*(\mathbf{x})), \tag{40}$$

where $(\mathbf{x}_*, \mathbf{w}_*(\mathbf{x}_*))$ is a Stackelberg equilibrium.

We then formulate unlearnable examples as a non-zero-sum Stackelberg formulation (Liu & Chawla, 2010; Huang et al., 2021; Yu et al., 2021; Fowl et al., 2021b;a; Sandoval-Segura et al., 2022; Fu et al., 2021):

$$\max_{\mathcal{D}_p} \mathcal{L}(\mathcal{D}_v, \mathbf{w}_*), \text{ s.t. } \mathbf{w}_* \in \arg\min_{\mathbf{w}} \mathcal{L}(\mathcal{D}_p, \mathbf{w}). \tag{41}$$

where $\mathcal{D}_p = \{(x_i + \sigma_i, y_i)\}_{i=1}^N$, $\sigma_i$ is the bounded sample-wise perturbation ($\|\sigma_i\|_p \leq \varepsilon_\sigma$), which can be generalized to class-wise perturbation (Huang et al., 2021) . Similar to indiscriminate data poisoning attacks, this primal formulation is difficult, as for the outer maximization problem, the dependence of $\mathcal{L}(\mathcal{D}_v, \mathbf{w}_*)$ on $\mathcal{D}_p$ or $\sigma$ is *indirectly* through the parameter $\mathbf{w}$ of the poisoned model.

However, in practice, we can perform a similar zero-sum reduction in Section 3 (Liu & Chawla, 2010):

$$\max_{\mathcal{D}_p} \min_{\mathbf{w}} \mathcal{L}(\mathcal{D}_p, \mathbf{w}). \tag{42}$$

We recall that in indiscriminate data poisoning attacks, such formulation is problematic as the attacker may simply perform well on poisoned points $\mathcal{D}_p$ but poorly on clean points. However, we would not encounter such a problem here as the perturbations are applied across the entire training set $\mathcal{D}_{tr}$.

Now we are ready to categorize existing algorithms on unlearnable examples:

- Error-Minimizing Noise (EMN) (Huang et al., 2021): By slightly modifying Equation (42) to

$$\min_{\mathbf{w}} \min_{\mathcal{D}_p} \mathcal{L}(\mathcal{D}_p, \mathbf{w}), \tag{43}$$

  Intuitively, Huang et al. (2021) construct the perturbation $\sigma$ to fool the model into learning a strong correlation between $\sigma$ and the labels.

Table 12: Indiscriminate data poisoning attacks: the attack accuracy/accuracy drop (%) and attack running time (hours) on CIFAR-10.

| Model | Clean | Label Flip | | EMN | | TGDA(ours) | |
|---|---|---|---|---|---|---|---|
| | Acc | Acc/Drop | Time | Acc/Drop | Time | Acc/Drop | Time |
| CNN | 69.44 | 68.99/0.45 | 0 hrs | 69.00/0.44 | 2.2 hrs | 65.15/**4.29** | 42 hrs |
| ResNet-18 | 94.95 | 94.79/0.16 | 0 hrs | 94.76/0.19 | 8.4 hrs | 89.41/**5.54** | 162 hrs |

Table 13: Unlearnable examples: model accuracy (%) under different unlearnable percentages on CIFAR-10 with ResNet-18 model. Percentage of unlearnable examples is defined as $\frac{|\mathcal{D}_p|}{|\mathcal{D}_{tr}+\mathcal{D}_p|}$.

| Method | 0% | 20% | 40% | 60% | 80% | 100% |
|---|---|---|---|---|---|---|
| EMN | 94.95 | 94.38 | 93.10 | 91.90 | 86.85 | 19.93 |
| TGDA | 94.95 | 93.22 | 92.80 | 91.85 | 85.77 | 16.65 |

- Robust Unlearnable Examples (Fu et al., 2021): Fu et al. (2021) further propose a min-min-max formulation following Equation (43):

$$\min_{\mathbf{w}} \ \min_{\sigma^u} \ \max_{\sigma^a} \mathcal{L}(\mathcal{D}'_p, \mathbf{w}), \tag{44}$$

where $\|\sigma_i^u\|_p \leq \varepsilon_u$ is the defensive perturbation, which is forced to be imperceptible; $\|\sigma_i^a\|_p \leq \varepsilon_a$ is the adversarial perturbation, which controls the robustness against adversarial training; $\mathcal{D}'_p = \{(x_i + \sigma_i^u + \sigma_i^a, y_i)\}_{i=1}^N$. Fu et al. (2021) find this formulation generates robust unlearnable examples against adversarial training.

- Adversarial poisoning (Error maximizing) (Fowl et al., 2021b): By freezing the follower entirely in Equation (42), Fowl et al. (2021b) propose to solve the maximization problem:

$$\max_{\mathcal{D}_p} \mathcal{L}(\mathcal{D}_p, \mathbf{w}), \tag{45}$$

such that it is similar to an adversarial example problem.

- Gradient Matching (Fowl et al., 2021a): Fowl et al. (2021a) solve the same maximization problem in Equation (45) and apply the gradient matching algorithm in Geiping et al. (2020) (see more details in Section 3).

## B.2 Comparison with Indiscriminate Data Poisoning Attacks

Despite their differences in problem formulation, it is possible to compare algorithms for unlearnable examples (we take EMN as an example here) with our TGDA attack. We identify two possible scenarios where we may fairly compare TGDA and EMN empirically:

- Indiscriminate Data Poisoning Attacks: for EMN, we first craft perturbations using the original algorithm. After the attack, we take $\mathcal{D}_p = \{(x_i + \delta_i), y_i\}_{i=1}^{\varepsilon N}$, recall $\varepsilon$ is the attack budget (or poison rate). Then, we follow our experimental protocol to perform the attack.
- Unlearnable Examples: for TGDA, we follow Equation (12) and perform the zero-sum reduction of TGDA to perturb the entire training set. Note that we only consider sample-wise perturbation across all experiments. Similar to our test protocol, we retrain the perturbed model and test the performance of the attack on the test set.

We report the experimental results in Table 12 and Table 13, where we observe that:

- For indiscriminate data poisoning attacks: In Table 12, although EMN is efficient, its attack efficacy is poor. Such poor performance is expected as the objective of EMN does not reflect the true influence of an attack on clean test data.
- For unlearnable examples: In Table 13, we observe that TGDA (after simplification) can be directly comparable with EMN, and the zero-sum simplification allows it to scale up to large models (i.e., ResNet) easily (training time for 100% unlearnable examples is 1.8 hours). However, the perturbation introduced by TGDA is not explicitly bounded.

## C    Other solvers than TGDA

We recall that in Section 3, we solve Equation (7) and approximate the calculation of $\frac{\partial \mathbf{w}_*}{\partial \mathcal{D}_p}$ using the total gradient descent ascent (TGDA) algorithm (Evtushenko, 1974; Fiez et al., 2020):

$$\mathbf{x}_{t+1} = \mathbf{x}_t + \eta_t \mathsf{D}_{\mathbf{x}} \ell(\mathbf{x}_t, \mathbf{w}_t), \tag{46}$$

$$\mathbf{w}_{t+1} = \mathbf{w}_t - \eta_t \nabla_{\mathbf{w}} f(\mathbf{x}_t, \mathbf{w}_t) \tag{47}$$

where $\mathsf{D}_{\mathbf{x}} := \nabla_{\mathbf{x}} \ell - \nabla_{\mathbf{w}\mathbf{x}} f \cdot \nabla_{\mathbf{w}\mathbf{w}}^{-1} f \cdot \nabla_{\mathbf{w}} \ell$ is the total derivative of $\ell$ with respect to $\mathbf{x}$.

Furthermore, it is possible to apply two other algorithms to solve Equation (7):

- Follow the ridge (FR) (Evtushenko, 1974; Wang et al., 2020):

$$\mathbf{x}_{t+1} = \mathbf{x}_t + \eta_t \mathsf{D}_{\mathbf{x}} \ell(\mathbf{x}_t, \tilde{\mathbf{w}}_t), \tag{48}$$

$$\mathbf{w}_{t+1} = \mathbf{w}_t - \eta_t \nabla_{\mathbf{w}} f(\mathbf{x}_t, \mathbf{w}_t) + \eta_t \nabla_{\mathbf{w}\mathbf{w}}^{-1} f \cdot \nabla_{\mathbf{x}\mathbf{w}} f \cdot \mathsf{D}_{\mathbf{x}} \ell(\mathbf{x}_t, \mathbf{w}_t), \tag{49}$$

- Gradient descent Newton (GDN) (Evtushenko, 1974; Zhang et al., 2021):

$$\mathbf{x}_{t+1} = \mathbf{x}_t + \eta_t \mathsf{D}_{\mathbf{x}} \ell(\mathbf{x}_t, \tilde{\mathbf{w}}_t), \tag{50}$$

$$\mathbf{w}_{t+1} = \mathbf{w}_t - \eta_t \nabla_{\mathbf{w}\mathbf{w}}^{-1} f \cdot \nabla_{\mathbf{w}} f(\mathbf{x}_t, \mathbf{w}_t), \tag{51}$$

Zhang et al. (2021) showed that both TGDA and FR are first-order approximations of GDN, despite having similar computational complexity of all three.

In our preliminary experiments, TGDA appears to be most effective which is why we chose it as our main algorithm.

