# OpenReview forum: "Indiscriminate Data Poisoning Attacks on Neural Networks"
_TMLR — Accepted by TMLR_

### Review · Reviewer_mWKi · 2022-10-16

**Summary Of Contributions:**

This paper studies indiscriminate data poisoning attacks on machine learning classifiers. The paper formulates data poisoning attacks and defenses as Stackelberg games, and based on the formulation, the paper proposes Total Gradient Descent Ascent (TGDA) poisoning attacks (that is a type of clean-label poisoning attacks). The proposed attack allows an adversary to efficiently generate a large number of poisoning samples. The paper evaluates the TGDA attacks against MNIST and CIFAR-10 classifiers and shows that the attacks achieve high accuracy drops while reducing the computational time for generating poisons.

**Audience:**

Yes

**Broader Impact Concerns:**

No concern.

**Claims And Evidence:**

No

**Requested Changes:**

1. Clarification of the motivation of this attack.
2. Clarification of the scientific advances this new formulation with Stackelberg makes.
3. Clarification of the technical advances this attack makes (other than just proposing an efficient attack)
4. Evaluation against many baseline indiscriminate attacks. (e.g., the paper claims the Min-max attack only works for convex models, but it seems not to be true. It does not guarantee the worst-case for non-convex models, but still works.)
5. Evaluation to back the "fairness" of evaluation claims.
6. Evaluation with the CIFAR10 models that achieve more than 90% accuracy.
7. Evaluation against potential defense mechanisms proposed in the community.
8. Comparison of the poisoning examples (clean-label) in terms of perturbations.

**Strengths And Weaknesses:**

Strengths

1. This paper studies the cost of indiscriminate poisoning attacks.
2. The paper proposes an efficient technique for generating a large number of poisons.
3. The paper is well-written and easy to follow.

Weaknesses

1. The motivation of this study is a bit weak.
2. The formulation of indiscriminate poisoning as Stackelberg games is somewhat unclear.
3. The techniques proposed for the TGDA attacks are less novel.
4. The experimental results are not sufficient to back up the paper’s claims.

Detailed comments

[Weak Motivation]

The TGDA attack reduces the computational costs of generating poisoning samples. However, I do not understand why the attacker may want to reduce the crafting costs. In indiscriminate poisoning, the objective is to reduce the accuracy of the target classifiers. Thus, it makes more sense to focus on reducing the number of poisoning samples the attacker injects or, with the same amount of poisoning samples, to increase an accuracy drop.


[Stackelberg Game]

I am a bit unclear about the benefits that the Stackelberg formulation brings. Typically, when we formulate a game between an attacker and a defender, we expect to explore and bind the limit of the attacker (e.g., similar to the work [1]). But, in this paper, the formulation seems to be only useful for making the poisoning sample generation faster. It raises a concern that we may not need the formulation to achieve the same goal—the TGDA attack.

[1] Steinhardt et al., Certified Defenses for Data Poisoning Attacks


[Technical Advances in the TGDA Attack]

The novelty is weak if the technical advance only reduces the time it takes to craft poisoning samples.

(1) I am again not sure why it’s a limitation of a prior work that they craft poisoning samples one by one, not in a batch. What will be the scenario where the attacker is under time pressure? In my opinion, the attacker can use as much time as they want as long as they can increase the accuracy drop of the target model after poisoning.

(2) It’s also a bit unclear whether the TGDA attack performs better than the others in terms of accuracy. The accuracy drop that this paper improves is from 1.52% (vs. label-flipping) /2.53 (vs. back-gradient) to 2.79% in LR trained on MNIST. I believe that this should be evaluated with multiple baselines; for example, even those two baselines, I think, will be weaker than the min-max attack proposed by Steinhardt et al.

[Weak Evaluations]

First, several benchmarks [1] evaluate the effectiveness of poisoning attacks. Thus, I believe that the claim that “there is a lack of systematic evaluation of poisoning attacks” should be toned down. At least, this should be only in the context of indiscriminate poisoning.

[1] Schwarzschild et al., Just How Toxic Is Data Poisoning? A Unified Benchmark for Backdoor and Data Poisoning Attacks

Second, the paper states that there is an unfair comparison between methods. However, I don’t see any evaluation that backs up this claim. It’s unclear (1) what “fairness” means and (2) whether (if there are some benchmark that exists) it is “unfair.”

Third, the clean models in CIFAR10 achieve a clean accuracy of 69%. Most recent models achieve at least 90% accuracy on CIFAR10; thus, I wouldn’t believe the impact of poisoning on those models. They could be amplified as the models used are too outdated and not good.

Fourth, in Sec 4.2, the attack process is designed to “thwart defenses,” but I couldn’t see the evaluation against most recent defenses (I only see one defense that seems way outdated than existing ones), such as Steinhardt et al.’s or some basic sanitization defenses. Some other work uses influence functions to identify poisoning samples. To make such claims, I think it’s necessary to evaluate the attack against existing defenses.

Fifth, it’s a clean-label poisoning attack. I think it is necessary to compare the magnitude of perturbations the TGDA attack introduces with the baseline attacks.


[Minor comments]

1. Backdoor attacks assume a completely different threat model (it assumes the attacker can manipulate the test-time data). I would remove the discussion about the backdoor attacks in the introduction and related work.
2. It may not be important to show the code is on GitHub in Table 1.
3. It’s unclear what “perfect autoencoder” in Figure 2 means.

---

> ### Author Response · Authors · 2022-10-17
> **Response to Reviewer mWKi**
>
> First, we want to sincerely thank Reviewer mWKi for the very rapid review. Here we want to clarify some questions, and we will update the response to the others (i.e., requested additional experiments) later.
> ****
> **Regarding comments that we can address now:**
>
> **(1) Weak Motivation and *(1)* in Technical Advances: why focus on efficiency?**
>
> In this paper we want to explore a fundamental question: can we perform indiscriminate attacks on neural networks?
>
> We find out it is usually infeasible as it is too expensive: e.g., in Table 2, it takes the Back-gradient attack 2153 hrs ($\approx$ 3 months) to poison a 3-layer CNN on MNIST. We agree that the attacker is not under any time pressure, but poor efficiency would significantly increase the computational cost and the difficulty of hyperparameter search, which potentially makes the attack weaker.
>
> Thus we believe the computational cost is the elephant in the room for poisoning neural networks, which we aim to overcome in the first place. Of course, efficacy is of vital importance. Thus we set TGDA attack as a baseline for indiscriminate attacks on neural networks and leave the efficacy improvement as future work.
>
> **(2) Stackelberg Game:**
>
> We understand the confusion. Here we want to clarify that the Stackelberg game serves two important functions:
> + it is an important tool for analyzing the order of the attacker and the defender (where previous works set the attacker as the leader by default). In Section 5.3, we show that who acts first matters: in Table 4, even without any defense strategy, the target model would be more robust if the defender acts one step ahead of the attacker.
> + The Stackelberg formulation (similar to the bilevel optimization formulation, e.g., in [1]) allows us to distinguish performance on the validation set and the poisoned points, thus revealing the zero-gradient issue.
>
> [1] Pang Wei Koh, Jacob Steinhardt, and Percy Liang. Stronger data poisoning attacks break data sanitization defenses.
>
> **(3) Technical advances:**
>
> Apart from efficiency, constructing multiple poisoned points at once allows more powerful coordination and hence could improve the performance of our attack. Here we conduct an experiment by applying TGDA to MNIST sequentially/in batches, where the results suggest that constructing poisoned points in batches also improves the effectiveness of an attack.
>
> | Method       | LR   | NN   | CNN  |
> |--------------|------|------|------|
> | Sequentially | 2.47 | 0.36 | 0.11 |
> | In batches     | 2.79 | 1.50 | 1.11 |
>
> **(4) Weak Evaluations**
>
> (a) the claim in the introduction should be tuned down: thank you for the suggestion, we will specify the context of indiscriminate poisoning and tune the claim down in the paper.
>
> (b) unfair comparison between methods: we understand the confusion, here we want to clarify that:
> + similar to Section 3 in [2], "unfair comparison" refers to the disparity in the attacker's knowledge, the attack formulation, and the dataset size in Table 1, and the inconsistent conclusions.
> + for better interpretation, we will rephrase the claim as “it is hard to compare these methods directly”.
>
> [2] Schwarzschild et al., Just How Toxic Is Data Poisoning? A Unified Benchmark for Backdoor and Data Poisoning Attacks
>
>
> **Minor comments**
> (a) We agree backdoor attacks are very different and we are happy to remove the discussion on backdoor attacks.
> (b) We will change the caption in Table 1.
> \(c\) A perfect autoencoder refers to an autoencoder with a very low reconstruction error, we will make it clear.
>
> ****
> **Regarding comments we are currently working on:**
>
> + *Evaluation against many baseline indiscriminate attacks*: we are conducting experiments and will add other baselines as requested.
> + *Evaluation with the CIFAR10 models that achieve more than 90% accuracy*: we agree that the simple model we used can be limited for analyzing the impact of poisoning. We are currently conducting experiments on more advanced models.
> + *Evaluation against potential defense mechanisms proposed in the community*: thank you for the advice, we will evaluate our attack against other existing defenses and add corresponding experiments.
> + *Comparison of the poisoning examples (clean-label) in terms of perturbations*: we agree such evaluation is valuable, we will add such experiments soon.
> ****
> **In summary**, we thank Reviewer mWKi again for the valuable suggestions, and we will try to finish the requested experiments as soon as possible and revise the paper accordingly. We will post an update once accomplished. In the meantime, we are happy to discuss any further concerns with the reviewer!

---

> > ### Author Response · Authors · 2022-11-16
> > **Update for Reviewer mWKi**
> >
> > Dear Reviewer mWKi,
> >
> >
> > We would like to thank you again for your rapid and insightful review. We are delighted to let you know that we have finished the requested experiments and revised the paper accordingly (marked in blue). In particular:
> >
> > **(1) Evaluation against many baseline indiscriminate attacks:**
> > Following your suggestion, we have conducted experiments on Min-max attack and i-Min-max (improved Min-max attack with target parameters). For logistic regression, we follow the algorithms in the original papers exactly. For neural networks, we ignore the convexity assumption and perform experiments. Note that for Min-max attack, its application on neural networks is similar to the zero-sum reduction of TGDA in Table 5.
> >
> > We report the results in Table 2 (updated version), where we observe that Min-max attack, due to its problematic formulation, does not work on neural networks. i-Min-max attack is effective against LR, but performs poorly on neural networks (presumably due to lack of convexity).
> >
> > **(2) Evaluation with the CIFAR-10 model that achieves more than 90% accuracy:**
> > To further improve the efficiency of our method, we propose an alternative approach: we split the dataset into 8 partitions and run TGDA separately on different GPUs. This approach is a heuristic practice that enables us to poison deeper models and we find it works well in practice: we are able to poison ResNet-18 on CIFAR-10 under 200 hours and cause a 5.54% accuracy drop (see updated Table 3). We have added relevant discussion in Section 5.
> >
> > **(3) Evaluation against stronger defense:**
> >
> > We examine the robustness of TGDA against two stronger defenses in Section 5.6 (b): the influence defense (remove points with the highest influence) in Koh & Liang 2017 (remove points with top singular value) and Sever defense in Diakonikolas et al. 2016. We observe in Table 11 that TGDA is robust against Influence defense, but its effectiveness is significantly reduced by Sever. Thus, we conclude Sever is potentially a good defense against TGDA, and it might require special design (e.g., an explicit constraint on the singular value) to break Sever sanitation.
> >
> >
> > **(4) Magnitude of perturbation:**
> >
> > We report the magnitude of perturbation in Figure 5 in terms of $l_{\infty}$ distance.
> >
> > Overall, we would like to thank Reviewer mWKi again for the great review, which has greatly helped us improve our presentation and discussion. We are looking forward to further discussing any additional questions with the reviewer!

---

> > > ### Comment · Reviewer_mWKi · 2022-11-21
> > > **Thank You for the Response**
> > >
> > > Most of my concerns are addressed from the authors' response. I am still unclear about whether we can answer this question in the end "can we perform indiscriminate attacks on neural networks?" because we only drop a few % on the benchmarks this paper tested. But, I could see the paper is on the way to answer this question; thus, in overall, I am positive about the revision.

---

> > > > ### Author Response · Authors · 2022-11-21
> > > > **Thank you for your positive comment!**
> > > >
> > > > Dear Reviewer mWKi,
> > > >
> > > > We appreciate your time and effort in reading our response and revision. And we are delighted to see that most of your concerns are addressed!

---

### Review · Reviewer_tVSb · 2022-10-29

**Summary Of Contributions:**

The paper summarizes a novel approach to data poisoning attacks on Neural Networks. The approach leverages TGDA(Total Gradient Descent Ascent) algorithm, which aims to address the attacker's objective. The paper talks about numerous approaches, such as how the defensive party can act to combat the attack. Their technique aims to be more effective and run on a higher magnitude. The experiment is based on the Stackelberg game, in which the attacker uses poisoned data to decrease accuracy while the defender optimizes the model using poisonous data.

**Audience:**

Yes

**Claims And Evidence:**

Yes

**Requested Changes:**

N/A

**Strengths And Weaknesses:**

Strengths:

1. All assumptions are explained well
2. The Stackelberg game is explained nicely
3. The different types of Data Poisoning attacks are explained in detail

---

> ### Author Response · Authors · 2022-11-16
> **Response to Reviewer tVSb**
>
> Dear Reviewer tVSb,
>
> We would like to thank you for your review and positive feedback! Please let us know if you have any additional questions.

---

### Review · Reviewer_TZgv · 2022-11-05

**Summary Of Contributions:**

This review is about TMLR submission 495. The submission "Indiscriminate Data Poisoning Attacks on Neural Networks" reviews and categorizes poisoning attacks on model availability and discuss a new attack. The main focus in this work is on model availability attacks (alternatively described as indiscriminate attacks) where the fraction of poisoned samples is small.  The attack is evaluated on MNIST and CIFAR10 with linear models and 3-layer neural networks.

**Audience:**

Yes

**Claims And Evidence:**

No

**Requested Changes:**

In summary, I would like for this submission to be revised, based on concerns discussed above. Most important for me, would be to:
* Improve the categorization of poisoning attacks in Sec. 3.
* Include attacks based on untrainable examples.
* Provide stronger attack scenarios in which the attack works well. The current attack budget is small, and so is attack success and this makes it hard to evaluate the work done in Sec. 3. and 4.
* Improve the engineering of the attack to make it feasible to attack deep neural networks, for example ResNets.

**Strengths And Weaknesses:**

In broad strokes I do think that this work engages an important topic. Model availability attacks with small budgets (which I will refer to as indiscriminate attacks following the author's notation from here on) against neural networks are a hard problem, where it is currently unclear if strong attacks are even possible, and I think it makes sense to approach this work with this framing in mind. I like the derivation of a new attack from bilevel literature via TGDA which I find well motivated.

Yet, to me at least, the current version of this submission falls short on delivering on the overall goal of providing a great attack. To put it briefly, I have concerns clustered around two main areas.

First, the submission promises a systematic analysis on poisoning deep neural networks, but reviews mostly Cinà et al. "Wild patterns reloaded: A survey of machine learning security against training data poisoning" and other surveys with fewer new insights than I would have hoped. Then, existing analysis based on Stackelberg games in the context of data poisoning is reviewed. Here, the submission opens several interesting questions that could go beyond previous treatments of Stackelberg games in poisoning (especially in terms of sequential Stackelberg games), but does not really act on this.
Second, the submission promises indiscriminate data poisoning attacks on modern neural networks, and deep neural networks, but the experimental results are not convincing enough to verify this, showing minimal performance degradations on 3-layer models at the cost of astronomical compute effort.

I'll provide a bullet-point list of more details below:

* The submission dismisses comparisons to attacks that perturb training data, but this seems an oversimplification to me. There is a decent number of works that investigate these "unlearnable examples", but these are also poisoning availability attacks. The only difference is that perturbations are constrained in input space, but not in number of poisoned examples.
It is straightforward to apply these attacks in scenarios where subsets of the training set are poisoned (or poisoned data is added) as analzyed in this work. This task of mixing clean and poisoned data is evaluated in several works on untrainable data, for example, in Sandoval-Segura "Autoregressive Perturbations for Data Poisoning" in Table 5. Methods tuned for untrainable data attacks might underperform in the 3% scenario investigated in this work, but the burden of proof there is on the submission to show this. A large range of attacks from this angle of work, especially error-minimizing and error-maximizing perturbations might still have a measurable impact when used in the scenario investigated in this work, especially when epsilon bounds are increased (or liffted entirely).
* More broadly, when setting out to categorize poison availability attacks, it seems surprising to dismiss these attacks, especially as many attacks can be cast as min-max optimization problems in line with the analysis based on zero-sum Stackelberg games, and works such as Fu et al. "Robust Unlearnable Examples: Protecting Data Privacy Against Adversarial Learning" can actually be understood as sequential Stackelberg games. I think it would be an insightful addition to include these works in the categorization.
* The authors discuss the use of TGDA as algorithm to approximate Eq.(5). It would be great to characterize previous attacks in relationship to TGDA. For example, gradient descent steps on the follower are also prominently discussed in related work. Huang et al., 2020, include a single descent step after evaluating K gradient steps on the leader [and average over many followers]. Muñoz-González et al.,2017, alternate single steps on both leader and follower (same as Koh 2017) and Geiping et al. 2021 "Witches' Brew: Industrial Scale Data Poisoning via Gradient Matching" approximate the leader dynamics via gradient matching and freeze the pretrained follower entirely. The update on the leader in Huang and Muñoz-González is actually not so different from the leader step arising from TGDA, as truncated backpropagation can converge to the same objective (see discussion in Shaban 2019, "Truncated back-propagation for bilevel optimization") and I think it would be interesting to conceptualize these works and others as instances of TGDA to strengthen the desired overview over poison attacks against neural networks, aside from only minor remarks in Sec. 4.2

* A central problem in previous work on data poisoning with formulations like TGDA is warm-starting, best summarized in Vicol 2022, "On Implicit Bias in Overparameterized Bilevel Optimization". It would be great if the submission could include a discussion of the problems discussed in Vicol 2022 for overparametrized models, in the context of this work.

* It is unclear to me why this attack is so expensive for these tiny machine learning models, especially on CIFAR-10. It would be great to at least include a breakdown of costs, but maybe even better would be to address this problem and provide a feasible algorithm that can be evaluated on modern neural networks. For example, have the authors considered the suggestions in Koh and Liang 2017 (Sec. 3), such as fast Pearlmutter approximations for the Hessian vector product? How many CG iterations are required?  In it's current state it feels hard to find this algorithm an "attack with improved efficiency" for neural networks.

* If I understand correctly, the attack of Muñoz-González is slow in Table 3 as poisons are created sequentially? While this arguably follows Muñoz-González to the letter, a more modern implementation could just compute poisoned data points in batches (or a single batch) by the same algorithm. I think this somewhat clouds the runtime comparison with the proposed approach.

* The submission claims that the proposed attack is very effective, but over all experiments (which really should include standard deviations), the effective drop is arguably small. The submission attempts to attack with a budget of 3% which really is a hard ask.  I think it would be much more convincing if aside from the 3% scenario, attacks would be plotted for a range of budgets, as done in Fig.4. and compared to other attacks on the full range.

* Speaking of Fig.4 (which I think shows MNIST? it would be great to mark this and also show the CIFAR variant), it is surprising that the proposed attack is not able to break the model even as epsilon goes to 1. Also just for clarification, at eps=1, is half the training set is filled with poisoned data, or is the entire dataset poisoned?

* Concerning the auto-encoder formulation to generate poisoned data: Overall I find this to be an interesting idea, but I do think it detracts somewhat from the central messaging in Sec. 3, as this component could be combined independently with any number of previous attacks. Essentially, this appears as a vague constraint on allowable images (it is, for example not clear what the worst-case image from this generator would look like and whether it still would be in-distribution to real data) that I would either formalize as part of the threat model or re-investigate. For example, another interpretation of the data Sec. 5.3 might be that this component regularizes poisoned data and helps to generate generalizable poisoning data points, with inconspiciousness more of a side effect.

* Table 7, what happens for m=n=1? Or variations with n=0?

* I liked the connection to seq. Stackelberg games briefly discussed in Sec. 5.3 and would have liked to see more analysis of this case. I do think that here again the critique of Radiya-Dixit et al. would be appropriate to discuss, namely that ultimately the defender always moves last in data poisoning.

Minor comments:
* The section 3 be strengthened by tying the discussion to older works analyzing adversarial games from this lens, for example
 Dalvie et al, 2004, "Adversarial classification" and Liu and Chawla, 2009 "A Game Theoretical Model for Adversarial Learning" and 2010 "Mining adversarial patterns via regularized loss minimization"
* Sec. 3.2., the submission notes that gradient descent ascent is infeasible due to zero gradients. While I agree with the sentiment, this seems false in general. It would be better to clarify the writing here, especially as the submission then turns to a decomposition of the total gradient (which also might be zero/undefined in large regions).
* CG is not a Hessian-free approach, I think the intention here was matrix-free?
* The defense section describes MaxUp as a good defense against adversarial examples. This is a statement that I would be careful with. In general, the only strong defense agaisnt adversarial examples is adversarial training.

---

> ### Author Response · Authors · 2022-11-16
> **Response to Reviewer TZgv (Part 1)**
>
> Firstly, we want to thank Reviewer TZgv for the very thorough review, which we find extremely insightful. We address the questions below, and we have also modified the paper accordingly (marked in blue).
>
> **(1) Comparison with unlearnable examples:**
>
> We agree that "unlearnable examples" also aim at decreasing the overall model accuracy and it is indeed feasible to experimentally compare our method with them. To make a relatively fair discussion, we set up two scenarios to compare with Error-Minimizing Noise (*EMN*):
> - **Indiscriminate attack by adding data** (our setting): Recall that we aim at decreasing the overall test accuracy, such that for
> (a) *TGDA*: we follow the formulation in our paper;
> (b) *EMN*: we first craft perturbations using the *EMN* algorithm. After the attack, we take $\mathcal{D}_p = \\{\(x_i+\delta_i\),y_i\\}_\{i=1\}^\{\epsilon N\}$, where $\epsilon$ is the attack budget (or poison rate). Such $\mathcal{D}_p$ is then used under our testing protocol.
> - **Unlearnable examples by perturbing the data** (setting in *EMN*): we recall that EMN aims at modifying the entire (or a subset of the) training set to decrease the overall test accuracy, such that for
>  (a) *TGDA*: here we can simplify the TGDA attack to the zero-sum setting in Equation (11) in our paper as we do not have clean data anymore.
>  (b) *EMN*: we follow the formulation in *EMN* for sample-wise perturbation.
>
> We conducted two sets of experiments and added them to the appendix in Tables 12 and 13. We briefly show them here:
>
> - **Indiscriminate attack:** here we choose $\epsilon=3\\%$ and obtain the following results on CIFAR-10 with the CNN architecture in our paper:
>
> | CIFAR-10   | Accuracy | Accuracy Drop | Running time |
> |------------|----------|---------------|--------------|
> | Clean      | 69.44%   | \             | 0            |
> | Label Flip | 68.99%   | 0.45%         | 0 hrs        |
> | TGDA       | 65.15%   | 4.29%         | 42 hrs      |
> | EMN        | 69.00%   | 0.44%         | 2.2 hrs      |
>
> We observe that although *EMN* is efficient, its attack efficacy is poor in our setting. Such performance is expected as the objective of *EMN* does not reflect the true influence of an attack on *clean* test data.
>
> - **Unlearnable examples:** here we choose to craft different percentages of unlearnable examples, following Table 2 in [18]. We obtain the following results on CIFAR-10 with ResNet-18.
>
>
>
> | CIFAR-10 | 0%     | 20%    | 40%    | 60%    | 80%    | 100%   |
> |--------------------|--------|--------|--------|--------|--------|--------|
> | EMN                | 94.95% | 94.38% | 93.10% | 91.90% | 86.85% | 19.93% |
> | TGDA               | 94.95% | 93.22% | 92.80% | 91.85% | 85.77% | 16.65% |
>
> We observe that TGDA (after simplification) can be directly comparable with EMN, and the zero-sum simplification allows it to scale up to large models (i.e., ResNet) easily (training time for 100% unlearnable examples is 1.8 hours). However, the perturbation introduced by TGDA is not explicitly bounded.
>
>
> **(2) Categorization in Section 3**
>
> We agree that "unlearnable examples", e.g., *EMN* and robust unlearnable examples can be understood as zero-sum sequential Stackelberg games, and such categorization would be insightful. However, the objective and application of "unlearnable examples", although similar, are still different from our main objective in Section 3 (i.e., Equation 7). We are a bit worried that placing such a discussion in Section 3 would somewhat raise confusion for the readers. Thus, we put all the relevant discussion in Appendix B for now, and we hope to further discuss with the reviewer if such discussion fits the main paper, and we are happy to modify it afterward.
>
> **(3) Characterize previous attacks in relationship to TGDA**:
>
> Thank you for the suggestion. Indeed, we summarize that:
> (a) Back-gradient attack can be understood as a k-step unrolled gradient descent algorithm. Additionally, TGDA can be treated as letting $k \rightarrow \infty$, and MetaPoison can be treated as an average over $M$ models. \
> (b) Witches' Brew applies gradient matching for a loss maximization problem (either for targeted attack setting or unlearnable examples), and can be indeed treated as taking 0 steps for the follower. \
> We have added relevant discussion in Section 3.2.
>
> Furthermore, we also identify other feasible solvers in the literature such as Follow the ridge (FR), and Gradient descent Newton (GDN) and add relevant discussion in Appendix C. We have empirically verified that FR and GDN, in their current implementation, are less effective for data poisoning attacks.

---

> > ### Author Response · Authors · 2022-11-16
> > **Response to Reviewer TZgv (Part 2)**
> >
> > **(4) Warm-start and cold-start**:
> >
> > Thank you for bringing this line of work to our attention! We find it extremely interesting to discuss the problem in the reference paper in the context of indiscriminate data poisoning attacks. Specifically, the methods we compare in this work all belong to the partial warm-start category for bilevel optimization. It is also possible to formulate the cold-start Stackelberg game for data poisoning. Specifically, we follow the reference paper such that in Algorithm 1, the follower update is modified to $\mathbf\{w\} \gets \mathbf\{w\}_\{pre\} - \beta \nabla_\{\mathbf\{w\}\} f(\theta, \mathbf\{w\})$. We report the results in Table 8 on MNIST in our modified draft and here:
> >
> > | Model | Clean | Partial Warm-start | Cold-start |
> > |-------|-------|--------------------|------------|
> > | LR    | 92.35 | 89.56/2.79         | 89.84/2.41 |
> > | NN    | 98.04 | 96.54/1.50         | 96.77/1.27 |
> > | CNN   | 99.13 | 98.02/1.11         | 98.33/0.80 |
> >
> > We observe that in indiscriminate data poisoning, partial warm-start is a better approach than cold-start overall, and our outer problem (autoencoder for generating poisoned points) does not appear to be highly over-parameterized.
> >
> > **(5) A more efficient attack**:
> >
> > (a) For CIFAR-10, TGDA takes more time to train than MNIST because the leader architecture applies the autoencoder and includes more parameters. Note that it requires 16 iterations of CG and we have added it in the modified draft.
> >
> > (b) We have checked the fast Perlmutter approximation approach, and it seems that a feasible PyTorch implementation would be applying the R-operator on the gradient in https://gist.github.com/apaszke/c7257ac04cb8debb82221764f6d117ad. However, this implementation only improves the efficiency a bit (saving 1/5 runtime approximately) in our case and still does not enable our algorithm to run on ResNet.
> >
> > \(c\) To further improve the efficiency of our method, we propose an alternative approach: we split the dataset into 8 partitions and run TGDA separately on different GPUs. This trick enables us to poison deeper models and we find it work well in practice: we are able to poison ResNet-18 on CIFAR-10 under 200 hours which causes a 5.54% accuracy drop (see updated Table 3). We have added relevant discussion in Section 5.
> >
> >
> >
> > **(6) Back-gradient in batches**:
> >
> > Yes, we follow the original paper and generate poisoned points sequentially. It is indeed possible to concatenate several poisoned points as input and feed into the Back-gradient algorithm. However, as the dimension of the input increases, the runtime for generating one point also increases. Overall we find this approach approximately takes a similar amount of time as the sequential generation.
> >
> >
> > **(7) More attack budget/Figure 4**:
> >
> > (a) We have added standard derivations in Table 2 and Table 3.
> >
> > (b) As some baseline attacks (e.g., i-Min-max and Back-gradient) do not scale to bigger $\epsilon$, it might be hard to compare with them. Thus, we have added a comparison with other methods on both MNIST and CIFAR-10 datasets.
> >
> > \(c\) For clarification, at $\epsilon=1$, only 50% of the training set is filled with poisoned data.
> >
> > (d) For $\epsilon=1$, it might not be too surprising that the attack cannot break the model: for example, in "Autoregressive Perturbations for Data Poisoning" Table 5, 40% poisoned data only introduces a 4.44% accuracy drop; in "Unlearnable Examples: Making Personal Data Unexploitable" Table 2, 60% poisoned data only introduces 3.05% accuracy drop.
> >
> > **(8) Auto-encoder formulation:**
> > Indeed, the autoencoder only introduces structural constraint during pretraining, not explicitly on the attack. To study the worst-case images, we print the $l_{\infty}$ distance between input and output space and choose the figures with the largest distance in Figure 5. We agree that autoencoder may help regularize the poisoned data.
> >
> > **(9) Table 7**:
> > We have added these extreme cases in Table 7, in particular, when m=n=1, the attack is less effective as the defender might not be fully trained to respond to the attack. When n=0, the attack is hardly effective at all as the target model is not retrained.
> >
> > **(10) Radiya-Dixit et al.:**
> > In Radiya-Dixit et al., the authors consider a scenario in "unlearnable examples" that the defender's model might evolve over time. Indeed, this scenario seems valid in our setting as well: such that the defender might choose a better machine learning model to accomplish the same task. Thus, we can link such a discussion to our transferability experiment in Section 5.5: we observe that more advanced models (e.g., CNN) seem to nullify the attack constructed for simpler models (e.g., NN, LR).
> >
> > However, the robustness of the attack over time might not be so important for our case: unlike "unlearnable examples", the poisoned data in our setting is not generated *once and for all*. Thus, even if the mode evolves, it is always possible to craft another attack.

---

> > > ### Author Response · Authors · 2022-11-16
> > > **Response to Reviewer TZgv (Part 3)**
> > >
> > > **(11) Minor comments:**
> > >
> > > (a) tying the discussion to older works: thank you for the suggestion, we have added the relevant discussion at the end of Section 3.1.
> > >
> > > (b) We have clarified the writing for the zero gradients claim.
> > >
> > > \(c\) We have modified it to a Hessian inverse-free approach.
> > >
> > > (d) We have modified the claim.
> > >
> > >
> > > Overall, we would like to thank Reviewer TZgv again for the great review, which has greatly helped us improve our presentation and discussion. We are looking forward to further discussing any additional questions with the reviewer!

---

> > > ### Comment · Reviewer_TZgv · 2022-11-27
> > > **Continued Response to Part 2**
> > >
> > > Interesting findings and revisions in this part as well. There are a few things where I should provide more context:
> > >
> > > > We have checked the fast Perlmutter approximation approach, and it seems that a feasible PyTorch implementation would be applying the R-operator on the gradient in https://gist.github.com/apaszke/c7257ac04cb8debb82221764f6d117ad.
> > >
> > > Let me provide additional context: What you are describing looks like the efficient way to compute a vector-Hessian product via automatic differentiation. However, this involves higher-order derivatives. A different trick, described in Pearlmutter 1994, "Fast Exact Multiplication by the Hessian", is to forego the higher-order directional derivative in the vector-hessian product entirely and to approximate the Hessian by forward differentiation of exact gradients. You'll see this strategy applied in modern deep learning, for example for the bilevel problem in Liu et al., "DARTS: Differentiable Architecture Search". Of course, whether this approximation is stable enough for the approximate inverse necessary for the TGDA step is entirely unclear.
> > >
> > > I am mainly providing this as additional context, you don't need to try this as well.
> > >
> > > > To further improve the efficiency of our method, we propose an alternative approach: we split the dataset into 8 partitions and run TGDA separately on different GPUs
> > >
> > > While this is a great idea, does this mean I should multiply numbers in Table 3 by 8, or is table 3 still showing GPU-hours per attack?
> > >
> > >
> > > > only 50% of the training set is filled with poisoned data
> > >
> > > Thanks, I was confused about this. I agree that the results make sense in this setting. It would probably be helpful to explain this again in the caption of Fig.4.

---

> > ### Comment · Reviewer_TZgv · 2022-11-27
> > **Continued Response to Part 1**
> >
> > Thank you for the detailed and extensive revision. I have some followup questions and comments.
> >
> > > We agree that "unlearnable examples", e.g., EMN and robust unlearnable examples can be understood as zero-sum sequential Stackelberg games, and such categorization would be insightful. However, the objective and application of "unlearnable examples", although similar, are still different from our main objective in Section 3 (i.e., Equation 7). We are a bit worried that placing such a discussion in Section 3 would somewhat raise confusion for the readers.
> >
> > Could you share more of your thoughts here? I've thought about this for a bit, but to be honest, it is not clear to me. The main objective is the same for both the indiscriminate poisoning described here and the poisoning attacks described as unlearnable examples, and objectives like Eq.(7) can also be found in work on unlearnable examples. I do agree that there is a minor notational difference between adding poisoned samples and modifying existing samples, but none of these unlearnable attacks make use of this difference in interesting ways. Further, the attack described in this submission also initialize poisoned points with existing data from the training set.
> >
> > Overall, it seems to me that this is just two communities calling the same problem by different names, and if that is the case, I think the categorization section in the submission would be strengthened by clearing this up. I think the added experimental results already speak to  this.

---

> ### Comment · Reviewer_TZgv · 2022-11-27
> **General Updated Response**
>
> Overall, I think this an interesting round of revisions for this work. I still have some questions concerning the categorization and relationship to unlearnable examples, as outlined above, but the revision has adressed concerns regarding attacks based on untrainable examples by experimenting with error-minimizing noise, and provided clarification for the attack strength in stronger scenarios.
>
> There has also been work to make this attack feasible for deep neural networks in this revision, which I find crucial to support the claims made in this work. I still would like the authors to walk back their claims a tiny bit surrounding the "efficiency" of this attack, but I now do agree that the attack is "feasible" against modern neural networks.

---

> > ### Author Response · Authors · 2022-11-28
> > **Further Response to Reviewer TZgv**
> >
> > Thank you for your updated response! We further address your questions below and we have modified the paper accordingly in blue.
> >
> > **(1) Unlearnable Examples**:
> >
> > (a) We agree with the reviewer that indiscriminate data poisoning and unlearnable examples are closely related, and we have rearranged the categorization on Pages 4 and 5 in Section 3.
> >
> > (b) The main differences are:
> > - existing unlearnable examples directly modify the training set (often all of it), while we consider adding epsilon percent to the clean training set. This seemingly minor difference can cause a significant difference in performance. For example, in Huang et al., Table 2 and Figure 2 suggest that the attack can only introduce a significant accuracy drop when the entire training set is modified. In comparison, adding poisoned points to a clean training set would never result in 100% modification in the augmented training set ;
> > - most works on unlearnable examples eventually modify their formulation into a zero-sum relaxation, possibly because that is the only viable way to modify the entire training set, while our experiments showed that zero-sum relaxation does not work very well when only a small portion of poisoned data is added to the clean data set.
> >
> >
> > \(c\) As our Fig 4 (as well as Table 2 and Figure 2 in Huang et al., and Table 5 in Sandoval-Segura et al.,) demonstrated, the percentage $\epsilon$ seems to play a significant role in the effectiveness of the attack. We agree that this seems like a minor notational difference, but we also think this difference may have often been overlooked.
> >
> > **(2) Additional context on Perlmutter approximation:**
> >
> > Thank you for providing the additional reference! This definitely clarifies our confusion and we are happy to try this approximation in our future work.
> >
> > **(3) Table 3:**
> >
> > Yes, we should multiply the numbers in Table 3 by 8 for a single GPU running time. We have clarified this in our modified draft.
> >
> > **(4) Figure 4:**
> >
> > Thank you. We have added the explanation in the caption.
> >
> > **(5) Efficiency:**
> >
> > We have tuned down the claim on the efficiency of the attack in our modified draft (Sections 1, 4, 5, and 6).
> >
> > Overall, we are delighted that many of your concerns are addressed. And we would like to thank you again for your very insightful review and updated response!

---

### Author Response · Authors · 2022-11-16
**Paper Update**

We would like to thank the action editor and all reviewers again for the effort and time they spend on our paper. We have revised the paper according to the reviewers' suggestions and marked the changes in blue. In summary, we have made the following changes:

- In Section 1 Introduction: we specify the analysis of indiscriminate data poisoning attacks to avoid overclaim.
- In Section 2 Background: we guide readers for more discussion on unlearnable examples in Appendix B.
- In Section 3 Total Gradient Descent Ascent Attack:
(a) we add discussions on previous works in Section 3.1.\
 (b) we guide readers for the categorization of unlearnable examples to Appendix B.\
 (c) we characterize the Back-gradient attack as the k-step unrolled gradient descent ascent algorithm and link it to the TGDA attack.\
(d) we add discussions on MetaPoison and Witche's Brew.
- In Section 5 Experiments: (a) we add Min-max and i-Min-max attack as baselines in MNIST experiments and relevant experiments in Table 2.\
 (b) we propose a training heuristic that enables us to run TGDA on CIFAR-10 for ResNet-18 and we include relevant experiments in Table 3.\
 \(c\) we add comparison with baseline methods and experiments on CIFAR-10 in Figure 4 for more insights on different $\epsilon$.  \
(d) we add instances for m=1, n=0, and m=n=1 in Table 7. \
(e) we add experiments on cold-start in Table 8.\
 (f) we add the magnitude of perturbation in Figure 5. \
(g) we add experiments against Influence defense and Sever defense in Table 11.

- Furthermore, we added Appendix B: Unlearnable Examples and Appendix C: Other solvers than TGDA.

---

### Decision · Action_Editors · 2022-12-12

**Recommendation:** Accept with minor revision

**Comment:**

This paper aims to answer the question of whether indiscriminate poisoning attacks are feasible on modern deep neural networks. It formulates indiscriminate poisoning attacks and defenses as Stackelberg games. The Stackelberg games formulation gives a little more insight into the attacker/defender scenario by showing that who acts first matters.

Reviewer tVSb was a solicit reviewer and they ended up giving a very terse review with little substance; therefore I put little weight on their recommendation. However, both reviewers mWKi and especially TZgv gave outstanding reviews that were exceedingly detailed and they engaged with the authors. Both of them recommended Leaning Accept.

A common point of confusion was whether the reduction of computation time from the TGDA attack was of importance. I think this is crucial to the point of this work that they push for computational feasibility rather than attack efficacy.

Overall, i think this paper's results are neither significant nor its methods novel. However, based on TMLR's acceptance criteria (https://www.jmlr.org/tmlr/acceptance-criteria.html) which says ...

"Crucially, it should not be used as a reason to reject work that isn't considered “significant” or “impactful” because it isn't achieving a new state-of-the-art on some benchmark. Nor should it form the basis for rejecting work on a method considered not “novel enough”, as novelty of the studied method is not a necessary criteria for acceptance. We explicitly avoid these terms (“significant”, “impactful”, “novel”), and focus instead on the notion of “interest”. If the authors make it clear that there is something to be learned by some researchers in their area from their work, then the criteria of interest is considered satisfied"

... I think it's reasonable to give this paper an accept because it provided clear, evidence-backed results, along with an interesting formulation, that are of interest to a portion (albeit limited) of the data poisoning community.

Authors: before camera-ready, please address the revisions from the reviewers. Further please revise the following:
Tone down the claim that other works crafted poison points sequentially. For example, both MetaPoison and Witches Brew crafted their poison points in batches and those poison points can coordinate with one another during the crafting.

**Audience:**

I think the appropriate audience of this paper is limited to those interested in security against model availability attacks. I hesitate to even say those interested in data poisoning attacks because I don't think the formulation and results of the paper apply to targeted poisoninig attacks. This paper mainly aims to answer of the question whether it's even feasible to do indiscriminate poisoning, which is a very different problem from targeted poisoning.

**Claims And Evidence:**

The initial draft contained claims that were overstated or insufficiently supported by evidence. These include that unlearnable examples are irrelevant and that the results on MNIST and CIFAR10 with a 3-layer neural network are a sufficient to be called a platform for modern neural networks. Further there was a lack of detailed comparisons. After revisions, the claims were properly toned down and important comparisons were included including warm-vs-cold start, batch vs sequential crafting, and EMN vs TGDA.

---

> ### Author Response · Authors · 2022-12-14
> **Upload of Camera Ready Revision**
>
> We would like to sincerely thank all reviewers and AE for their insightful comments! We have uploaded the camera-ready revision which addresses the reviewers' concerns. Furthermore, we have toned down the claim that other works crafted poisoned points sequentially in the abstract, Section 1, and Section 4.

---

> ### Author Response · Authors · 2023-01-02
> **Thanks!**
>
> Thanks Ronny for overseeing our paper! Just wanted to add in my two cents: you're right that the data poisoning community seems to be dominated by people working on targeted poisoning and backdoor attacks. At this point, I think we understand those problems rather well. Our motivation was to explicitly work on something *different* from where all the attention has been recently been dedicated. Indeed, the indiscriminate setting has been rather underappreciated, and as our paper highlights, there are still rather substantial fundamental questions which are unresolved, and thus deserving of much more attention. We thus hope the data poisoning community chooses to investigate these problems more!
>
> Gautam Kamath